# Randomized Dimensionality Reduction for Euclidean Maximization and Diversity Measures

**Jie Gao** [1]  **Rajesh Jayaram** [2]  **Benedikt Kolbe** [3]  **Shay Sapir** [4]  **Chris Schwiegelshohn** [5]  **Sandeep Silwal** [6]
**Erik Waingarten** [7]

## Abstract

Randomized dimensionality reduction is a widely-used algorithmic technique for speeding up large-scale Euclidean optimization problems. In this paper, we study dimension reduction for a variety of maximization problems, including max-matching, max-spanning tree, max TSP, as well as various measures for dataset diversity. For these problems, we show that the effect of dimension reduction is intimately tied to the *doubling dimension* $\lambda_X$ of the underlying dataset $X$—a quantity measuring intrinsic dimensionality of point sets. Specifically, we prove that a target dimension of $O(\lambda_X)$ suffices to approximately preserve the value of any near-optimal solution, which we also show is necessary for some of these problems. This is in contrast to classical dimension reduction results, whose dependence increases with the dataset size $|X|$. We also provide empirical results validating the quality of solutions found in the projected space, as well as speedups due to dimensionality reduction.

## 1. Introduction

Dimensionality reduction is a key technique in data science and machine learning that allows one to transform a dataset of $n$ points in a $d$-dimensional space into a dataset where each point now lies in a much smaller $t$-dimensional space. This reduction in dimensionality occurs while (approximately) preserving the dataset's essential properties. The benefits of such reduction are apparent—the total storage of a $d$-dimensional dataset is $O(nd)$. After dimension reduction, the total storage is $O(nt)$.

Another aspect is running time. After a dimension reduction, we may replace dependencies on $d$ with a dependency on $t$. This seemingly addresses the curse of dimensionality which is used to describe the phenomenon that algorithms scale poorly in high dimensions. Initially this seems very promising, as many problems admit a polynomial time approximation scheme (PTAS) in a low-dimensional setting (Bartal & Gottlieb, 2021; Cevallos et al., 2019; Kolliopoulos & Rao, 2007). However, these algorithms typically have severe dependency on the dimension $d$ (e.g., could be doubly exponential). We are thus interested in reducing the dimension as much as possible, ideally even to constants.

Problems for which efficient algorithms are known to exist in low-dimensions, but are not known to exist in high-dimensions, include network design problems, such as the traveling salesperson (Arora, 1997; Kolliopoulos & Rao, 2007; Shenmaier, 2022), and diversity maximization problems (Cevallos et al., 2019; 2018), both of which are staples of data analysis and operations research. However, this discrepancy between low- and high-dimensional data exists primarily in the worst case. One can consider a beyond worst-case analysis by attempting to identify tractable inputs to the problem which are not necessarily low dimensional (Awasthi et al., 2010; Ostrovsky et al., 2012). One way of utilizing such properties is to design a dimension reduction to uncover such structures that might not be easily used in the original high dimension, see (Awasthi & Sheffet, 2012; Cohen-Addad & Schwiegelshohn, 2017).

Dimensionality reduction techniques can be broadly categorized into two distinct types: data-dependent and data-

---

*Authors listed alphabetically. [1]Department of Computer Science, Rutgers University, Piscataway, NJ, USA [2]Google Research, NYC, USA [3]Hausdorff Center for Mathematics, Lamarr Institute for Machine Learning and Artificial Intelligence, University of Bonn, Germany [4]Department of Computer Science, Weizmann Institute of Science, Israel [5]Department of Computer Science, Aarhus University, Denmark [6]Department of Computer Sciences, University of Wisconsin-Madison, USA [7]University of Pennsylvania, Philadelphia, USA. Correspondence to: Jie Gao <jg1555@cs.rutgers.edu>, Rajesh Jayaram <rkjayaram@google.com>, Benedikt Kolbe <bkolbe@uni-bonn.de>, Shay Sapir <shay.sapir@weizmann.ac.il>, Chris Schwiegelshohn <cschwiegelshohn@gmail.com>, Sandeep Silwal <silwal@cs.wisc.edu>, Erik Waingarten <ewaingar@seas.upenn.edu>.

*Proceedings of the 42$^{nd}$ International Conference on Machine Learning*, Vancouver, Canada. PMLR 267, 2025. Copyright 2025 by the author(s).

oblivious. Data-dependent techniques, like Principal Components Analysis (PCA) and t-SNE (van der Maaten & Hinton, 2008), aim to produce "tailor-made" low-dimensional representations via (frequently) computationally-intensive procedures (possibly with quadratic running times). The benefit of being data-dependent is that one can capture the non-worst-case nature of "real world" datasets; the downside is that data-dependent methods are computationally expensive, see (Boutsidis et al., 2009; Cohen et al., 2015; Feldman et al., 2020) for applications of PCA to clustering and subspace approximation.

In data-oblivious dimension reduction, the goal is to produce—without even looking at the dataset—a dimensionality-reducing map from $\mathbb{R}^d$ to $\mathbb{R}^t$. These are considerably more "lightweight" and simple to apply and indeed vital for streaming and distributed models, where the applied dimensionality reduction method is restricted to a small portion of the entire input. Indeed, the Johnson-Lindenstrauss (JL) transform (Johnson & Lindenstrauss, 1984), which can be realized by a $t \times d$ matrix of independent $\mathcal{N}(0, 1/t)$ entries (Indyk & Motwani, 1998; Dasgupta & Gupta, 2003), is among the most popular examples. Despite being data-oblivious, JL transforms are surprisingly powerful. They preserve pairwise distances, and also center-based clustering (Makarychev et al., 2019; Izzo et al., 2021), linkage clustering (Narayanan et al., 2021), subspaces (Charikar & Waingarten, 2022), and cuts (Chen et al., 2023), to name a few applications.

Among these works, (Indyk & Naor, 2007; Narayanan et al., 2021) stand out in relating the performance of the JL transform to the *doubling dimension* of the input dataset, incorporating elements of data-dependent and data-oblivious dimension reduction. On the one hand, the dimensionality-reducing map is still a JL transform; it is data-oblivious, simple to apply, store and communicate. On the other hand, the analysis is data-dependent—it ties the performance of the map to a data-dependent quantity, the *doubling dimension*, which captures the intrinsic dimensionality and non-worst-case nature of high-dimensional datasets. Formally, the doubling dimension is the smallest number $\lambda$ such that for every ball of radius $r$, the input points in that ball can be covered by $2^\lambda$ balls of radius $r/2$ (Gupta et al., 2003) (see also (Clarkson, 1999)). For arbitrary worst-case instances, $\lambda \leq \log n$, as we can always cover $n$ points with $2^{\log n} = n$ balls for any choice of radius $r$. For any given instance, however, the quantity may be substantially smaller. Perhaps surprisingly, it is possible to prove data-dependent bounds on the performance of data-oblivious dimension reduction, deriving the benefits of both.

So far, this type of analysis (incorporating the doubling dimension to data-oblivious dimension reduction) has only been done for facility location (Narayanan et al., 2021;

Huang et al., 2024), single-linkage clustering (Narayanan et al., 2021), approximate nearest neighbors (Indyk & Naor, 2007) and $k$-center (Jiang et al., 2024).

We ask: *For which problems can we design a data-oblivious dimension reduction with data-dependent target dimension? Can we identify a large collection of related geometric problems which admit such dimensionality reduction?*

## 1.1. Our Contributions

In this paper, we identify a large class of problems, with a special focus on *network design* (maximum weight matching, max TSP, max spanning tree) and *diversity maximization problems* (subgraph diversity maximization), for which we are able to obtain a data-oblivious dimension reduction with data-dependent target dimension. Maximum weight matching, max TSP and max spanning tree are natural network design problems where the weights describe profit or rewards on the edges. Maximum diversity problems arise in many application settings, from facility location to network analysis, and have been a topic of study in both computer science (Indyk et al., 2014) and operational research (Martí et al., 2022). In general, such problems focus on identifying a subset of elements from a given set such that a diversity/distance measure is maximized. Different problems in this family choose ways to measure the overall diversity measure. Possibly the most notable problem is the dispersion maximization, which maximizes the smallest pairwise distance of the selected elements. But other measures of using the sum of edge lengths in a subgraph (clique, star, cycle, tree, matching, and pseudoforest) have also been studied (Indyk et al., 2014). Taking diversity into account in data analysis is also gaining traction in machine learning (Gong et al., 2018) especially in generative models (Eigenschink et al., 2023; Naeem et al., 2020) and active learning (Melville & Mooney, 2004; Yang et al., 2015).

Given input dataset with doubling dimension $\lambda$, we show that reducing the dimension to roughly $t = O(\lambda)$, using a matrix of i.i.d. Gaussians drawn from $N(0, \frac{1}{t})$, preserves solutions up to a factor of $1 + \varepsilon$ on the problems in Table 1 found *by any algorithm* with high probability. Furthermore, we also show that the dependence on $\lambda$ in the target dimension is tight for all of these problems — there are datasets for which reducing the dimension below $\Omega(\lambda)$ introduces large errors.

A particularly nice feature of our bounds is that the guarantee applies to *all* candidate solutions. We may thus use *any algorithm* in the low dimensional space in post processing. We are only aware of a comparable result for nearest neighbor search (Indyk & Naor, 2007). Previous work in this direction only preserved the value of the optimum (for MST (Narayanan et al., 2021)) or restricted the candidate solutions (for facility location (Narayanan et al., 2021; Huang

| Problem | Target Dimension | Reference |
|---|---|---|
| Max weight matching | $O(\varepsilon^{-2}\lambda\log\frac{1}{\varepsilon})$ | Thm 2.1 |
| | $\Omega(\lambda)$ | Thm 3.1 |
| Max TSP | $O(\varepsilon^{-2}\lambda\log\frac{1}{\varepsilon})$ | Thm 2.1 |
| | $\Omega(\lambda)$ | Thm 3.5 |
| Max $k$-hypermatching | $O(\varepsilon^{-2}\lambda k^2\log\frac{k}{\varepsilon})$ | Thm 2.3 |
| Max spanning tree | $O(\varepsilon^{-2}\lambda\log\frac{1}{\varepsilon})$ | Thm B.5 |
| | $\Omega(\lambda)$ | Thm B.6 |
| Max $k$-coverage | $O(\varepsilon^{-2}\lambda\log\frac{1}{\varepsilon})$ | Cor B.4 |
| | $\Omega(\lambda)$ | Thm B.6 |
| Subgraph diversity | $O(\frac{1}{\varepsilon^2}(\lambda\log\frac{1}{\varepsilon}+\log k))$ | Cor A.2 |
| | $\Omega(\lambda)$ | Thm A.3 |

*Table 1.* Summary of results for dimension reduction with $1+\varepsilon$ distortion using a matrix of i.i.d. Gaussians. The doubling dimension of the input dataset is denoted by $\lambda$.

et al., 2024) and $k$-center (Jiang et al., 2024)).

**Threshold Phenomena** As another contribution, we identify an intriguing sharp threshold phenomenon for maximum matching, and we believe a similar phenomena also occurs for some of the other problems (e.g., maximum TSP). While reducing the dimension to $O(\lambda)$ (or $O(\log n)$ in general) provides a $(1+\varepsilon)$-approximation of maximum matching for any constant $\varepsilon > 0$; there are datasets such that reducing the dimension slightly below this bar causes a distortion of at least $\sqrt{2}$ (by Theorem 3.1). Curiously, the approximation factor remains at most $\sqrt{2}$ all the way down to target dimension $O(1)$ (see Theorem 4.1). We are not aware of any previous work establishing similar phenomena.

### 1.2. Preliminaries

For simplicity, we only consider random linear maps defined by a matrix of Gaussians as in many prior works.

**Definition 1.1.** A Gaussian JL map is a $t \times d$ matrix with i.i.d. entries drawn from $N(0, \frac{1}{t})$.

The following is a crucial lemma in our analysis, which bounds the expansion of balls under a Gaussian JL map.

**Lemma 1.2** (Lemma 4.2 in (Indyk & Naor, 2007))**.** Let $X \subset B(\vec{0}, 1)$ be a subset of the Euclidean unit ball. Then there are universal constants $c, C > 0$ such that for $t > C \cdot \mathrm{ddim}(X) + 1$, $D > 1$, and a Gaussian JL map $G \in \mathbb{R}^{t \times d}$, $\Pr(\exists x \in X, \|Gx\| > D) \le e^{-ctD^2}$.

## 2. Maximum Matching and Max-TSP for Doubling Sets

In this section, we prove dimensionality reduction results for maximum matching and maximum traveling salesman problem. Denote by $\mathrm{opt}_{\text{max-match}}$ and $\mathrm{opt}_{\text{max-tsp}}$ the optimum

of maximum matching and maximum TSP, respectively.

**Theorem 2.1.** *Let $0 < \epsilon < 1$ and $d, \lambda \in \mathbb{N}$, and Gaussian JL map $G \in \mathbb{R}^{t \times d}$ with suitable $t = O(\epsilon^{-2}\lambda\log\frac{1}{\epsilon})$. Then for every set $P \subset \mathbb{R}^d$ with $\mathrm{ddim}(P) = \lambda$, with probability at least 2/3, every $(1 + \varepsilon)$-approximate solution for $\mathrm{opt}_{\text{max-match}}(G(P))$ or $\mathrm{opt}_{\text{max-tsp}}(G(P))$ is a $(1 + O(\varepsilon))$-approximate solution to $\mathrm{opt}_{\text{max-match}}(P)$ or $\mathrm{opt}_{\text{max-tsp}}(P)$, respectively. Moreover, $O(1)$-approximate solutions are preserved as well.*

The proof of the theorem is by the following lemma.

**Lemma 2.2.** *Let $0 < \epsilon < 1$ and $d, \lambda \in \mathbb{N}$, and a Gaussian JL map $G \in \mathbb{R}^{d \times t}$ with suitable $t = O(\epsilon^{-2}\lambda\log\frac{1}{\epsilon})$. Then, for every set $P \subset \mathbb{R}^d$ with $\mathrm{ddim}(P) = \lambda$, with probability $\ge 9/10$, we have, for every matching $M$ of $P$,*

$$\sum_{\{p,q\}\in M}\|Gp-Gq\| \le \sum_{\{p,q\}\in M}\|p-q\|+\varepsilon\cdot\mathrm{opt}_{\text{max-match}}(P).$$

*Proof of Theorem 2.1.* Notice that for both problems, $\mathrm{opt}(G(P)) \ge (1-\varepsilon)\,\mathrm{opt}(P)$ w.h.p. by the following. Consider an optimal solution $S$ in $P$ (where $S$ is either a perfect matching or a tour of $P$). We have

$$\mathrm{opt}(G(P)) \ge \mathrm{cost}(G(S)) = \sum_{\{p,q\}\in S}\|Gp - Gq\|,$$

which is w.h.p. $\ge (1-\varepsilon)\sum_{\{p,q\}\in S}\|p - q\|$ (e.g., by analyzing the expectation and applying Markov's inequality). Therefore, the result for maximum matching is immediate from Lemma 2.2.

For maximum TSP, consider a tour $S$ of $G(P)$. It decomposes to three matchings $M_1, M_2, M_3$ (in fact, if $n$ is even then $M_3$ is the empty set, and otherwise it contains a single edge). Thus,

$$\mathrm{cost}(G(S)) = \sum_{i=1}^{3}\sum_{\{p,q\}\in M_i}\|Gp - Gq\|$$

$$\le \sum_{i=1}^{3}\sum_{\{p,q\}\in M_i}\|p - q\| + 3\varepsilon\cdot\mathrm{opt}_{\text{max-match}}(P)$$

$$= \mathrm{cost}(S) + 3\varepsilon\cdot\mathrm{opt}_{\text{max-tsp}}(P).$$

Rescaling $\varepsilon$ concludes the proof. □

Our result for max-matching generalizes to hypermatching, where edges in the matching are replaced with $k$-hyperedges (e.g., triangles). Formally, in the max $k$-hypermatching problem, $n$ is a multiple of $k$, and the goal is to partition $P$ into $n/k$ sets $S_1, \ldots, S_{n/k}$, each of size $k$, to maximize $\sum_{i=1}^{n/k}\sum_{p,q\in S_i}\|p - q\|$.

**Theorem 2.3.** *Let $0 < \epsilon < 1$ and $d, \lambda, k \in \mathbb{N}$, and Gaussian JL map $G \in \mathbb{R}^{t \times d}$ with suitable $t = O(\epsilon^{-2} k^2 \lambda \log \frac{k}{\epsilon})$. Then for every set $P \subset \mathbb{R}^d$ with $\mathrm{ddim}(P) = \lambda$, with probability at least $2/3$, a $(1+\varepsilon)$-approximation to the maximum $k$-hypermatching of $G(P)$ is a $(1 + O(\varepsilon))$-approximation to the maximum $k$-hypermatching of $P$.*

The proof is essentially by picking $\varepsilon' = \frac{\varepsilon}{k}$ and applying Theorem 2.1. The proof is provided in Appendix C.

### 2.1. Proving Lemma 2.2.

A key observation in our proof is the following.

**Lemma 2.4.** *Let $M$ be a maximum matching of a dataset $P$ with radius $r$. There exists a ball of radius $\frac{r}{2}$ that contains every $(p, q) \in M$ for which $\|p - q\| \leq \frac{r}{4}$.*

*Proof.* Let $(p, q), (x, y) \in M$ such that $\|p - q\| \leq \frac{r}{4}$ and $\|x - y\| \leq \frac{r}{4}$. We claim that $x, y \in B(p, \frac{r}{2})$, from which the lemma follows.

The proof is by contradiction. Suppose $\|p - x\| > \frac{r}{2}$ (wlog, the same arguments hold for $y$), then there is a matching $M'$ with cost larger than $M$ — it is the same as $M$, but with a minor change, match the pairs $(p, x), (q, y)$ (i.e., the pairs are switched). Indeed, the difference between the cost of $M'$ and $M$ is

$$\sum_{(a,b) \in M'} \|a - b\| - \sum_{(a,b) \in M} \|a - b\|$$
$$= \|p - x\| + \|q - y\| - (\|p - q\| + \|x - y\|)$$
$$> \frac{r}{2} - 2 \cdot \frac{r}{4} = 0,$$

and $M'$ has a larger cost, a contradiction. $\square$

*Proof of Lemma 2.2.* Let $M$ be a maximum matching of $P$. Denote by $r$ the radius of $P$. We construct a sequence of balls, such that intuitively, for every $(p, q) \in M$, there exists a ball $B$ in this sequence that contains $p, q$ and $\|p - q\| = \Omega(\mathrm{radius}(B))$.

The construction is as follows. Let $B_0$ be the minimum enclosing ball of $P$. Let $P_1$ be the set of points whose distance from their match in $M$ is $\leq \frac{r}{4}$ and $B_1$ be a ball of radius $\frac{r}{2}$ given by Lemma 2.4. In particular, $P_1 \subset B_1$. Let $M_1$ be the matching $M$ induced on $P_1$. Clearly, $M_1$ is a maximum matching of $P_1$ (whose radius $\leq \frac{r}{2}$), so we can apply Lemma 2.4 again, getting a ball $B_2$ and $P_2$ with radius smaller by a factor of 2. Proceed in this manner, so for every $i \in \mathbb{Z}_+$, $B_i$ has radius $r_i = \frac{r}{2^i}$. By construction,

$$\forall i \in \mathbb{Z}_+, \ \forall (p, q) \in M_i \setminus M_{i+1}, \qquad \|p - q\| \geq \frac{r_i}{4}. \quad (1)$$

For every $p \in P$, denote the last level that contains $p$ by $i_p$, i.e., the maximum $i$ such that $p \in P_i \setminus P_{i+1}$, and denote the radius of that level by $r_p$.

*Claim* 2.5. $\sum_{p \in P} r_p \leq 8 \, \mathrm{opt}(P)$.

*Proof of Claim 2.5.*

$$\sum_{p \in P} r_p = \sum_{i=0}^{\infty} \sum_{(p,q) \in M_i \setminus M_{i+1}} (r_p + r_q)$$
$$\leq \sum_{i=0}^{\infty} \sum_{(p,q) \in M_i \setminus M_{i+1}} 8\|p - q\| = 8 \, \mathrm{opt}(P),$$

where the inequality is by Equation (1). $\square$

We will use a helpful probability statement.

*Claim* 2.6 (Claim C.2 of (Makarychev et al., 2019)). There exists a universal constant $c > 0$ such that for all $t < d$, and for a Gaussian JL map $G \in \mathbb{R}^{t \times d}$,

$$\forall x \in \mathbb{R}^d, r > 0, \qquad \Pr(\|Gx\| > (1 + r)\|x\|) \leq e^{-cr^2 t}.$$

We now analyze the error contributed by each ball. For every $i \in \mathbb{Z}_+$, let $N_i$ be an $\epsilon r_i$-net of $P \cap B_i$. By standard arguments, it has size $\leq N := (2/\epsilon)^{\mathrm{ddim}(P)}$ (see e.g., (Gupta et al., 2003)). For suitable target dimension $t = O(\varepsilon^{-2} \log N) = O(\epsilon^{-2} \mathrm{ddim}(P) \log \frac{1}{\epsilon})$, the following holds for a matrix $G \in \mathbb{R}^{t \times d}$ of i.i.d. $N(0, \frac{1}{t})$: For every $\alpha > 1, x \in \mathbb{R}^d$, by Claim 2.6,

$$\Pr(\|Gx\| > (1+\epsilon\alpha)\|x\|) \leq \exp(-\Omega(\alpha^2 \varepsilon^2 t)) \leq N^{-2} 2^{-\alpha^2}.$$

By a union bound over all pairs $x, y \in N_i$, with probability $\geq 1 - 2^{-\alpha^2}$,

$$\forall x, y \in N_i, \qquad \|Gx - Gy\| \leq (1 + \epsilon\alpha)\|x - y\|. \quad (2)$$

Additionally, for every $y \in N_i$, we have by Lemma 1.2 that with probability $\geq 1 - e^{-ct\alpha^2}$,

$$\forall p \in P \cap B(y, \epsilon r_i), \qquad \|G(p - y)\| \leq \alpha\varepsilon r_i. \quad (3)$$

Thus, by a union bound, with probability $\geq 1 - e^{-ct\alpha^2}|N_i| \geq 1 - 2^{-\alpha^2}$ the above hold for all $y \in N_i$. For every $i$, denote by $\alpha_i$ the smallest $\alpha \geq 1$ for which Equations (2) and (3) hold. Therefore, for every $j \in \mathbb{N}$,

$$\Pr(\alpha_i > 2^{j-1})$$
$$\leq \Pr(\exists x, y \in N_i, \|Gx - Gy\| > (1 + \varepsilon 2^{j-1})\|x - y\|)$$
$$+ \Pr(\exists y \in N_i, p \in P \cap B(y, \varepsilon r_i), \|Gp - Gy\| > 2^{j-1}\varepsilon r_i)$$
$$\leq 2 \cdot 2^{-4^{j-1}}.$$

Hence, $\mathbf{E}\alpha_i \leq \sum_{j=0}^{\infty} 2^j \Pr(\alpha_i \geq 2^{j-1}) \leq \sum_{j=0}^{\infty} 2^j 2 \cdot 2^{-4^{j-1}} = O(1)$.

Next, we consider a random variable that captures the error accumulated by all the balls together. Denote $X := \sum_{i=0}^{\infty} \alpha_i |P_i| r_i$. We have,

$$\mathbf{E}X = \sum_{i=0}^{\infty} \mathbf{E}\alpha_i |P_i| r_i = O(\sum_{i=0}^{\infty} |P_i| r_i)$$
$$= O(\sum_{p \in P} r_p) = O(\mathrm{opt}(P)). \tag{4}$$

By Markov's inequality, w.p. $> 9/10$, we have $X = O(\mathrm{opt}(P))$. Assume this event holds.

Finally, we are ready to prove the lemma. Let $M'$ be a matching of $P$. We bound the cost of this matching on $G(P)$. For every $p \in P$, consider level $i_p$ (recall, it is the last level containing $p$). For every $(p, q) \in M'$, assume wlog that $i_p \leq i_q$ (so $B_{i_q} \subseteq B_{i_p}$). Denote by $y_p \in N_{i_p}$ the nearest net-point to $p$ in that level, and by $y_{qp} \in N_{i_p}$ the nearest net-point to $q$ in the same level $i_p$. In particular, $p \in B(y_p, \varepsilon r_p)$ and $q \in B(y_{qp}, \epsilon r_p)$. By triangle inequality,

$$\sum_{(p,q) \in M'} \|Gp - Gq\|$$
$$\leq \sum_{(p,q) \in M'} \|Gp - Gy_p\| + \|Gq - Gy_{qp}\| + \|Gy_p - Gy_{qp}\|$$

by Equations (2) and (3),

$$\leq \sum_{(p,q) \in M'} 2\varepsilon \alpha_{i_p} r_p + (1 + \varepsilon \alpha_{i_p}) \|y_p - y_{qp}\|$$

since $y_p, y_{qp}$ are in a ball of radius $r_p$, and by triangle inequality,

$$\leq \sum_{(p,q) \in M'} 4\varepsilon \alpha_{i_p} r_p + \|y_p - p\| + \|y_{pq} - q\| + \|p - q\|$$
$$\leq \sum_{(p,q) \in M'} 4\varepsilon \alpha_{i_p} r_p + 2\epsilon r_p + \|p - q\|$$

and by Equation (4),

$$= O(\varepsilon \, \mathrm{opt}(P)) + \sum_{(p,q) \in M'} \|p - q\|.$$

This concludes the proof of Lemma 2.2. □

# 3. $\sqrt{2}$ Approximation Lower Bound for Max Matching and Max-TSP

The aim of this section is to derive a lower bound on the dimension necessary for the maximum matching (and max TSP) of a projected set of points to yield a better than $\sqrt{2}$ approximation for the optimal cost of the original point set. Omitted proofs can be found in Appendix D.

**Theorem 3.1.** *Let $\varepsilon \in (0,1)$, $n = \Omega(1/\varepsilon^2)$ be even and $P = \{e_1, \cdots, e_n\}$ be the standard basis vectors in $\mathbb{R}^n$. Let $G$ be a Gaussian JL map onto $t$ dimensions. If $\frac{C \log(1/\varepsilon)}{\varepsilon^2} \leq t \leq \frac{\log(n)}{C \log(1/\varepsilon)}$ for a sufficiently large constant $C$, then with probability $\geq 1 - \exp(-\Omega(n\varepsilon^2)) - 1/n^{50}$, we have*

$$\mathrm{opt}_{\text{max-match}}(G(P)) \geq \left(\sqrt{2} - \varepsilon\right) \cdot \mathrm{opt}_{\text{max-match}}(P).$$

The proof is by demonstrating a matching with large cost in the target dimension. Note that $\mathrm{opt}_{\text{max-match}}(P) = \sqrt{2} \cdot n/2$, since all pairwise distances equal $\sqrt{2}$. After dimension reduction, the points are i.i.d. distributed $N(0, \frac{1}{t} I_t)$, hence are roughly mapped to be uniform on the unit sphere. Thus, we expect points to have an antipodal match at distance 2, so the matching would have size roughly $2 \cdot n/2$. However, this holds on average, and it is unclear how to construct a matching. We resolve this as follows.

**Lemma 3.2.** *Let $\varepsilon \in (0,1), t \geq C\frac{\log(1/\varepsilon)}{\varepsilon^2}$ for a sufficiently large constant $C > 0$. Let $x_1, \cdots, x_n$ be i.i.d. draws from $\mathcal{N}(0, I_t)$. With probability $\geq 1 - \exp(-\Omega(n\varepsilon^2))$, at least $1 - \varepsilon$ fraction of the $x_i$ satisfy $\|x_i\|_2 / \sqrt{t} \in 1 \pm \varepsilon$.*

**Lemma 3.3.** *Let $\varepsilon \in (0,1), n = \Omega(\varepsilon^{-2})$ and $\frac{C \log(1/\varepsilon)}{\varepsilon^2} \leq t \leq \frac{\log(n)}{C \log(1/\varepsilon)}$ for a sufficiently large constant $C > 0$. Let $x_1, \cdots, x_{n/2}$ and $y_1, \cdots, y_{n/2}$ be i.i.d. draws from $\mathcal{N}(0, I_t)$. Consider a bipartite graph $H = (A, B)$ with bipartition $A = \{x_1, \cdots, x_{n/2}\}, B = \{y_1, \cdots, y_{n/2}\}$. Put an edge between $x_i \in A$ and $y_j \in B$ if $\|x_i + y_j\|_2 \leq \varepsilon\sqrt{t}$. With failure probability at most $1/n^{99}$, we have*

$$|\deg(x_1) - \mathbf{E}[\deg(x_1)]| \leq \varepsilon \mathbf{E}[\deg(x_1)].$$

**Lemma 3.4.** *Consider the setting of Lemma 3.3. If $C$ is a sufficiently large constant then $H$ has a matching of size $\geq n/2(1 - \varepsilon)$ with probability at least $1 - 1/n^{50}$.*

For the proof of Lemma 3.4, recall the linear programming relaxation of the maximum matching problem on a bipartite graph $H = (A \cup B, E)$: $\max \sum_{(i,j) \in E} z_{ij}$ subject to

$$0 \leq z_{ij} \leq 1, \forall (i,j) \in E$$
$$\sum_{j \in B} z_{ij} \leq 1, \forall i \in A, \quad \sum_{i \in A} z_{ij} \leq 1, \forall j \in B.$$

It is a classic fact that any feasible (possibly fractional) solution to the above LP can be rounded to an integral solution with cost at least as large as the fractional solution (even in polynomial time but we only need existence here) (Chakrabarty). Using this, we can prove Lemma Lemma 3.4, whose full proof is deferred to Appendix D. The main idea is that we set each $z_{ij}$ to be $((1 + \varepsilon)\Delta)^{-1}$ where $\Delta = \mathbf{E}[\deg(x_1)]$. The capacity constraints are satisfied since the capacity constraint for vertex $x_i$ sums to

$\deg(x_i)/((1+\varepsilon)\Delta) \leq 1$ by our degree concentration of Lemma 3.3. One can check that this assignment also leads to a large matching.

*Proof of Theorem 3.1.* We will prove the theorem by demonstrating a matching with large objective value in the projected dimension. As noted earlier, $\text{opt}_{\text{max-match}}(P) = \sqrt{2} \cdot n/2$. Furthermore, $Ge_i$ for $1 \leq i \leq n$ are all independent standard Gaussian vectors, scaled down by $1/\sqrt{d}$. Partition the $Ge_i$'s into two disjoint sets of size $n/2$. Lemma 3.4 implies that (with high probability) we can find a matching of size $\geq n/2(1 - \varepsilon/100)$ among points across these two sets, satisfying that the vertices on every edge of the matching sum to a vector with $\ell_2$ norm at most $\varepsilon/100$. Furthermore, Lemma 3.2 implies that at least a $(1 - \varepsilon/1000)$ fraction of vectors in both sets satisfy that their $\ell_2$ norms are $1 \pm \varepsilon/100$ (again with high probability). By a union bound, at least a $(1 - \varepsilon/10)$ fraction of the edges in the matching have *both* endpoints with $\ell_2$ norms in $1 \pm \varepsilon$. That is, we can find $n/2(1 - \varepsilon/10)$ pairs of vectors (in the projected space) such that every pair $(x, y)$ satisfies (1) $\|x + y\|_2 \leq \varepsilon/100$, (2) $\|x\|_2, \|y\|_2 \in 1 \pm \varepsilon$. By the reverse triangle inequality, this implies

$$\|x - y\|_2 \geq |2 \cdot \|x\|_2 - \|x + y\|_2|$$
$$= |2(1 \pm \varepsilon) - \|x + y\|_2| \geq 2 - \Omega(\varepsilon).$$

Thus, with high probability (more precisely with probability at least $1 - \exp\left(-\Omega(n\varepsilon^2) - 1/n^{50}\right)$ which follows from combining the failure probabilities of Lemmas 3.4 and 3.2), we have demonstrated that a matching of weight at least $\frac{n}{2}(1 - \varepsilon) \cdot (2 - \Omega(\varepsilon)) \geq \text{OPT}(P) \cdot (\sqrt{2} - \Omega(\varepsilon))$ exists in the projected space. □

We prove a similar lower bound as in Theorem 3.1 for the maximum weight TSP problem. The theorem below follows from carefully combining Theorem 3.1 on small subsets of the input dataset to form a large cost tour.

**Theorem 3.5.** *Let $\varepsilon \in (0, 1)$, $n = \Omega(1/\varepsilon^3)$ and $P = \{e_1, \cdots, e_n\}$ be the standard basis vectors in $\mathbb{R}^n$. Let $G$ be a Gaussian JL map onto $t$ dimensions. If $\frac{C \log(1/\varepsilon)}{\varepsilon^2} \leq t \leq \frac{\log(n)}{C \log(1/\varepsilon)}$ for a sufficiently large constant $C$, then with probability $\geq 1 - \exp(-\Omega(n\varepsilon^3)) - 1/n^{25}$, we have*

$$\text{opt}_{\text{max-tsp}}(GP) \geq \left(\sqrt{2} - \varepsilon\right) \cdot \text{opt}_{\text{max-tsp}}(P).$$

## 4. $\sqrt{2}$-Approximation for Maximum Matching

We consider dimension reduction for maximum matching without the doubling assumption. We provide a similar result of 2-approximation for maximum TSP in Appendix E.

**Theorem 4.1.** *Let $\epsilon > 0$ and $d \in \mathbb{N}$. There is a random linear map $g : \mathbb{R}^d \to \mathbb{R}$, such that for every set $P \subset \mathbb{R}^d$ of even size, the ratio $\mathbf{E}[\text{opt}_{\text{max-match}}(g(P))]/\text{opt}_{\text{max-match}}(P)$ is in $[1, \sqrt{2}]$. Moreover, for a Gaussian JL map $G \in \mathbb{R}^{t \times d}$ with $t = O(\varepsilon^{-2})$, we have with probability at least $2/3$ that $\text{opt}_{\text{max-match}}(G(P))$ is a $(\sqrt{2} + \varepsilon)$-approximation of $\text{opt}_{\text{max-match}}(P)$.*

In order to prove this theorem, we use the following lemma, which may be of independent interest. For a point set $P \subset \mathbb{R}^d$, the 1-median of $P$ finds a point $c \in \mathbb{R}^d$ that minimizes $\sum_{p \in P} \|c - p\|$.

**Lemma 4.2.** *For every set $P \subset \mathbb{R}^d$, the cost of the 1-median of $P$ is a $\sqrt{2}$-approximation to $\text{opt}_{\text{max-match}}(P)$. That is,*

$$\max_{\text{matching } M \text{ on } P} \sum_{\{u,v\} \in M} \|u - v\|$$
$$\leq \min_{c \in \mathbb{R}^d} \sum_{p \in P} \|c - p\| \leq \sqrt{2} \max_{\text{matching } M \text{ on } P} \sum_{\{u,v\} \in M} \|u - v\|.$$

*Proof of Theorem 4.1.* Let $g \in \mathbb{R}^d$ be a vector of i.i.d. Gaussians distributed $N(0, \pi/2)$. One could verify that for every $x \in \mathbb{R}^d$, $\mathbf{E}[|g^\top x|] = \|x\|$. Denote by $c^*$ a point that realizes the 1-median of $P$, and by $M^*$ a maximum matching of $P$. The following holds.

$$\mathbf{E}\left[\max_{\text{matching M}} \sum_{\{u,v\} \in M} |g^\top u - g^\top v|\right]$$
$$\geq \mathbf{E}\left[\sum_{\{u,v\} \in M^*} |g^\top u - g^\top v|\right] = \sum_{\{u,v\} \in M^*} \|u - v\|.$$

This immediately gives the lower bound of the claim. On the other hand, we have

$$\mathbf{E}\left[\min_{c \in \mathbb{R}} \sum_{p \in P} |g^\top p - c|\right] \leq \mathbf{E}\left[\sum_{p \in P} |g^\top p - g^\top c^*|\right]$$
$$= \sum_{p \in P} \|p - c^*\| = \min_{c \in \mathbb{R}^d} \sum_{p \in P} \|p - c\|.$$

Together with Lemma 4.2, we now have

$$\mathbf{E}\left[\max_{\text{matching M}} \sum_{\{u,v\} \in M} |g^\top u - g^\top v|\right]$$
$$\leq \mathbf{E}\left[\min_{c \in \mathbb{R}} \sum_{p \in P} |g^\top p - c|\right]$$
$$\leq \min_{c \in \mathbb{R}^d} \sum_{p \in P} \|p - c\| \leq \sqrt{2} \sum_{\{u,v\} \in M^*} \|u - v\|.$$

The "moreover" part follows by standard analysis of the expectation and variance of Chi-Squared distribution. □

To prove Lemma 4.2, we use results on *Tverberg graphs*. A weighted graph $G = (V, E, w)$ is called a Tverberg graph

if its vertex set $V$ is a subset of $\mathbb{R}^d$, the weight of every edge $e = \{u, v\} \in E$ is $w(e) = \|v - u\|_2$, and if one places a diametrical ball $B(uv) = B(\frac{v+u}{2}, \frac{\|v-u\|_2}{2})$ for every edge $e = \{u, v\} \in E$, then these balls intersect, i.e., $\bigcap_{\{v,u\} \in E} B(vu) \neq \emptyset$. We will use the following result by (Pirahmad et al., 2024).

**Lemma 4.3** ((Pirahmad et al., 2024)). *For every pointset $P \subset \mathbb{R}^d$ of even size, there exists a perfect matching that is a Tverberg graph.*

*Proof of Lemma 4.2.* By triangle inequality,

$$\min_{c \in \mathbb{R}^d} \sum_{p \in P} \|c - p\| \geq \max_{\text{matching } M} \sum_{\{u,v\} \in M} \|u - v\|.$$

It thus remains to prove the other direction.

Denote by $M^*$ a matching given by Lemma 4.3, and by $p^*$ a point in $\bigcap_{\{v,u\} \in M^*} B(vu)$. Let $\{v, u\} \in M^*$. Therefore,

$$\|v - p^*\| + \|u - p^*\| \leq \sqrt{2}\sqrt{\|v - p^*\|^2 + \|u - p^*\|^2}$$
$$\leq \sqrt{2}\|v - u\|,$$

where the first inequality is due to Cauchy–Schwarz inequality and the second inequality is because $p^*$ is inside the ball with $uv$ as diameter and the angle $up^*v$ is not acute, as illustrated in Figure 1. Thus,

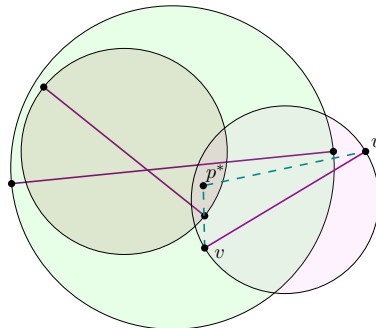

*Figure 1.* A Tverberg graph and a point in the intersection of all disks. The angle $up^*v$ is not acute.

$$\min_{c \in \mathbb{R}^d} \sum_{p \in P} \|c - p\| \leq \sum_{p \in P} \|p^* - p\|$$
$$\leq \sum_{\{u,v\} \in M^*} \sqrt{2}\|u - v\| \leq \sqrt{2} \max_{\text{matching } M} \sum_{\{u,v\} \in M} \|u - v\|,$$

concluding the proof. $\square$

## 5. Empirical Evaluation

We complement our theoretical results with an empirical evaluation. Our goal is to convey two key messages: first,

randomized dimensionality reduction can be very effective in reducing the runtime of geometric computation in high dimensions. This is not surprising and indeed, many empirical and theoretical findings in the literature already validate this hypothesis (this is the entire basis of the long line of work on the JL lemma). Our contribution is simply showing that the expected speedups are also possible for the various geometric problems we study, including many well-studied measures of dataset diversity maximization (to the best of our knowledge, we are the first to consider these problems under the lens of dimensionality reduction).

The second and the more salient point we demonstrate is that the effects of doubling dimension is an *empirically observable* phenomenon which can be quantitatively measured. In particular, seemingly similar datasets living in the same ambient dimension, but with different 'intrinsic dimensionality,' (which we theoretically formalize in terms of the doubling dimension) can behave wildly differently with respect to dimensionality reduction. As expected, datasets with lower intrinsic dimension can be projected to significantly smaller dimension while still preserving 'relevant' (with respect to the problem at hand) geometric information.

**Datasets** We use two real and one synthetic dataset. For each dataset, we create a 'low-intrinsic dimension' version and a 'high-intrinsic dimension' version. Both versions will live in the *same* ambient dimension and ostensibly they will be very similar, however, they will approximately correspond to versions of the dataset with low/high doubling dimension. We demonstrate that the 'low-dimensional' versions have much more desirable behavior with respect to dimensionality reduction.

- Dataset 1: MNIST. We select 1000 randomly chosen images from the MNIST dataset (dimension 784 with entries normalized to be in $[0, 1]$), restricted to the digit 2. We fixed the digit 2 since it has been studied in prior works as an example of a dataset with low-doubling dimension (Tenenbaum et al., 2000; Narayanan et al., 2021). This dataset is labeled **MNIST '2'** in our figures. For the 'high-intrinsic dimension' version, we add i.i.d. Gaussian entries to all coordinates (note the noise is of the same scale as the original entries). This dataset is labeled **MNIST '2' + Gaussian** in our figures.

- Dataset 2: CIFAR-embeddings. We use penultimate layer embeddings of pre-trained ResNet models (Yu et al., 2021; Backurs et al., 2024) in $\mathbb{R}^{6144}$. We use 1000 randomly chosen embeddings. These are the 'high-intrinsic dimension' version of the dataset. This is labeled **Cifar-High** in our figures. For the 'low-intrinsic dimension', we only keep the first 128 dimensions of these embeddings, but rotate the resulting points to also lie in $\mathbb{R}^{6144}$ using a random orthogonal matrix. This procedure simulates 'hiding'

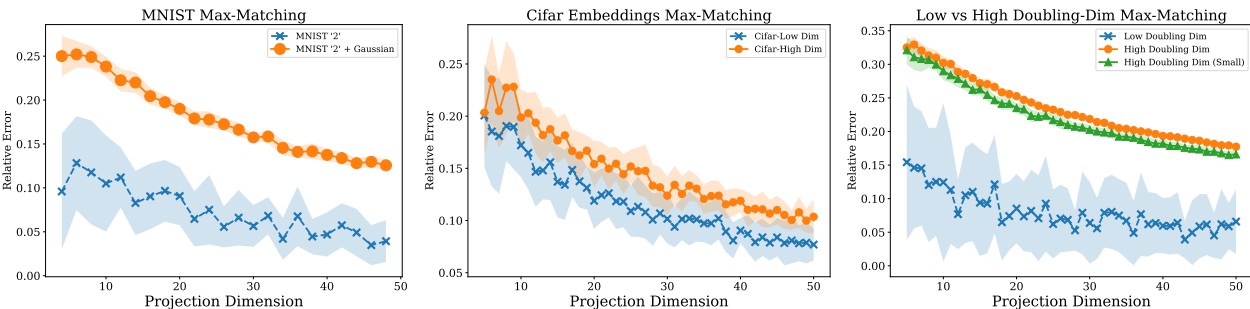

Figure 2. Relative error versus projection dimension for maximum-matching.

truly low-dimensional data in high-dimension. Note that this is not so obvious a-priori, since the rotation ensures the vectors are dense vectors in $\mathbb{R}^{6144}$. This is labeled **Cifar-Low** in our figures.

- Dataset 3: Synthetic. The 'high-intrinsic dimension' version of this dataset consists of the $n$ basis vectors in $\mathbb{R}^n$. This is labeled **High Doubling Dim** in our figures. The 'low-intrinsic dimension' version of the dataset is the cumulative sums of the basis vectors, i.e. $e_1, e_1 + e_2, \ldots, e_1 + \ldots + e_n$. These two versions appear to be similar, but it can be shown that their respective doubling dimensions are $\Omega(\log n)$ and $O(1)$, respectively. They were also considered in the experiments of (Narayanan et al., 2021). This is labeled **Low-Doubling Dim** in our figures. We also consider a 'Small' version of the 'high-intrinsic dimension' dataset, where we just take the first $n/2$ basis vectors. This is labeled **High Doubling Dim (Small)** in our figures. We set $n = 1000$ in our experiments.

Note that throughout our figures, orange represents the 'high-intrinsic dimensionality' versions of our datasets and blue represents the 'low-intrinsic dimensionality' versions. Green is only used for our synthetic dataset, where we also have a third version where we take half of the basis vectors (the 'high-intrinsic dimensionality' case) as a separate dataset version.

**Optimization Problems** We focus our attention to three representative problems among the many that we study. We pick maximum weight matching, remote-clique, and max $k$-coverage. We selected these three since they are representative of our main theoretical results. For maximum matching, we experiment with the bipartite version by simply dividing our dataset in half. This allows us to use an efficient exact solver using SciPy (Virtanen et al., 2020). The other two problems correspond to well-studied dataset diversity measures, many known to be NP-Hard to optimize exactly (Erkut, 1990; Chandra & Halldórsson, 2001), so we use a greedy algorithm as a proxy for finding the optimum.

Our greedy strategy builds the size $k$ set in remote-clique and max $k$-coverage iteratively, by picking the best choice at every step. Such greedy heuristics have been extensively used in the applied literature on diversity maximization (see (Mahabadi & Narayanan, 2023) and references within), and demonstrate that they produce solutions quite close to the ground truth for real world datasets (Mahabadi & Trajanovski, 2024; Gollapudi et al., 2024).

For each problem, we compute a solution (using the aforementioned algorithms) in the original dimension. Then we reduce the dimension of our datasets, varying the target dimension, and run the same computation on the projected data. We then compare the values of the two solutions found and measure the relative error between them. In all of our results we plot the average over at least 20 independent trials and one standard deviation error is shaded. We implement our algorithms using Python 3.9.7 on an M1 MacbookPro with 32GB of RAM.

**Results: Effect of High vs Low Doubling Dimension**
Our figures empirically demonstrate that the 'high-intrinsic dimensionality' versions of all of our datasets require a much larger projection dimension to achieve the same relative error compared to the 'low-intrinsic dimensionality' versions of our datasets. This is displayed in Figure 2 for the max-matching problem and the same qualitative results can be observed for our other two problems (see Figures 3 to 6 in Appendix F). As a sanity check, the relative errors are decreasing in the projection dimension.

In Figure 2, we see for the MNIST dataset that the relative error achieved by projecting into 50 dimensions for the 'high-intrinsic dimensionality' version (orange curve) can already be achieved at a much smaller projection dimension (approximately 20) for the 'low-intrinsic dimensionality' version (blue curve). The same qualitative phenomena can be observed across datasets and problems that we study.

We would like to especially highlight the third plot in Figure 2. There, the orange and green curves are virtually identical, and both are much higher than the blue curve. However,

the blue curve corresponds to a point set with *twice* the cardinality than the point set for the green curve! Thus, for the max-matching problem (and also for the two other problems we experiment on), the performance of dimensionality reduction is not determined by the size of the point set, as the naive application of the JL lemma suggests. Rather for these problems, the doubling dimension is the quantity of interest, as characterized by our theory.

**Results: Speedups Due to Dimensionality Reduction**
Lastly, we mention that, as expected, performing the computation in the projected space is much more efficient. Our Figures (see Figure 2 for maximum matching, Figures 3 and 4 for remote clique, and Figures 5 and 6 for max $k$-coverage) already demonstrate that we can achieve small relative error for high dimensional datasets, by projecting them to dimensions as low as 20. Indeed, across datasets and problems, optimizing for our three different objectives in the original ambient dimension, versus optimizing for them after projecting onto dimension 20 leads to speed ups of up to **two orders** of magnitude. The speedups for computing the objective in the projected space (taking into account the time required to perform the dimensionality reduction) is shown in Table 2.

## Acknowledgements

This research was initiated during the Workshop "Massive Data Models and Computational Geometry" held at the University of Bonn in September 2024 and funded by the DFG, German Research Foundation, through EXC 2047 Hausdorff Center for Mathematics and FOR 5361: KI-FOR Algorithmic Data Analytics for Geodesy (AlgoForGe). We thank the organizers and participants for the stimulating environment that inspired this research.

Jie Gao would like to acknowledge NSF support through IIS-22298766, DMS-2220271, DMS-2311064, CRCNS-2207440, CCF-2208663 and CCF-2118953. Chris Schwiegelshohn the support by the Independent Research Fund Denmark (DFF) under a Sapere Aude Research Leader grant No 1051-00106B. Erik Waingarten thanks the NSF support under grant CCF-2337993.

## Impact Statement

This paper presents work whose goal is to advance the field of Machine Learning. There are many potential societal consequences of our work, none which we feel must be specifically highlighted here. Our work is of theoretical nature.

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

# A. Remote Subgraph Diversity Measures

In this section, we consider remote subgraph problems, which include many well studied diversity measures. The goal is to find a subset $S \subset P$ of size $k$ that maximizes a certain diversity measure of $S$. The diversity measures we consider satisfy the following.

**(P)** Given a set of points $S$, compute the minimum/maximum of the sum of edges over a subgraph family, where all elements of the graph family have the same number of edges.

**Theorem A.1.** *Let $0 < \epsilon < 1$ and $d, \lambda, k \in \mathbb{N}$, and Gaussian JL map $G \in \mathbb{R}^{t \times d}$ with suitable $t = O(\epsilon^{-2}(\lambda \log \frac{1}{\epsilon} + \log k))$. Then, for every set $P \subset \mathbb{R}^d$ with $\mathrm{ddim}(P) = \lambda$, with probability at least $2/3$, the following holds. Let a diversity measure $\mathrm{div}$ and denote by $\pi$ the problem of finding a subset $S \subset P$ of size $k$ that maximizes $\mathrm{div}(S)$; if $\mathrm{div}$ satisfies Property (P), then every $(1 + \varepsilon)$-approximate solution for $\pi(G(P))$ is a $(1 + O(\varepsilon))$-approximate solution to $\pi(P)$.*

**Corollary A.2.** *The theorem above holds for the following diversity measures of the set $S$:*

- *Remote-edge: $\min_{p,q \in S} \|p - q\|$.*

- *Remote-clique: $\sum_{p,q \in S} \|p - q\|$.*

- *Remote-tree: weight of a minimum spanning tree of $S$.*

- *Remote-star: $\min_{p \in S} \sum_{q \in S \setminus p} \|p - q\|$.*

- *Remote-cycle: length of a minimum TSP tour of $S$.*

- *Remote-matching: weight of a minimum perfect matching of $S$.*

- *Remote-pseudoforest: $\sum_{p \in S} \mathrm{dist}(p, S \setminus p)$.*

The dependence on the doubling dimension is tight, as can be seen by the following.

**Theorem A.3.** *Let $n \in N$, there exists a set $P$ of $n$ points, such that for a Gaussian JL map $G$ onto dimension $t$. Let $\pi$ be a problem as defined in Theorem A.1. If $\pi$ is defined for $k = 2$, then $\pi(G(P)) \geq \Omega(\sqrt{\frac{\log n}{t}}) \cdot \pi(P)$ with high probability.*

This theorem can be seen as a direct corollary of a similar statement for the diameter of the dataset; one can find a proof in the full version of (Jiang et al., 2024).

**Lemma A.4.** *Under the same conditions of Theorem A.3, the diameter is distorted by factor $\Omega(\sqrt{\frac{\log n}{t}})$ with high probability.*

*Proof of Theorem A.3.* For $k = 2$, the problems for which Theorem A.1 applies are identical to computing the diameter. Applying Lemma A.4 concludes the proof. $\square$

## A.1. Proof of Theorem A.1

The proof of Theorem A.1 uses the following two lemmas. For a point set $P$, the $k$-center problem finds a subset $S \subseteq P$ with $|S| = k$ such that every point in $P$ is within distance $r$ from some point in $S$, with the goal of minimizing the radius $r$. We denote by $KC_k(P)$ the radius in the optimal $k$-center solution. The following lemma is well-known. See e.g. Lemma 2 in (Abbar et al., 2013).

**Lemma A.5.** *Let $d, k \in \mathbb{N}$ and a point-set $P \subset \mathbb{R}^d$. There exists a set $S \subset P$ of $k$ points such that for every $x, y \in S, x \neq y$, we have $\|x - y\| \geq KC_{k-1}(P)$.*

The next lemma is implicit in (Jiang et al., 2024). We provide a proof at the end of this section for completeness.

**Lemma A.6.** *Let $0 < \epsilon < 1$ and $d, k \in \mathbb{N}$, and a Gaussian JL map $G \in \mathbb{R}^{t \times d}$ with suitable $t = O(\epsilon^{-2}(\lambda \log \frac{1}{\epsilon} + \log k))$. Then, for every set $P \subset \mathbb{R}^d$ with $\mathrm{ddim}(P) = \lambda$, with probability at least $2/3$, for all $x_1, x_2 \in P$,*

$$\|Gx_1 - Gx_2\| \in (1 \pm \epsilon)\|x_1 - x_2\| \pm \epsilon KC_k(P).$$

*Proof of Theorem A.1.* Consider target dimension as in Lemma A.6, and suppose the event therein holds. Consider a remote subgraph problem $\pi$ with diversity measure $\mathrm{div}$ satisfying (P), and suppose that given a $k$-point set, $\mathrm{div}$'s value is a sum over a set of $l$ edges. Assume that $\mathrm{div}$ is a minimization problem, as is the case for all problems in Corollary A.2. The proof regarding maximization problems is by the same arguments.

Consider a set $\tilde{S}$ of size $k$ given by Lemma A.5. Let $\tilde{E}$ be a set of $l$ pairs of points (edges) in $\tilde{S}$ that realize $\mathrm{div}(\tilde{S})$. By Lemma A.5, $\mathrm{div}(\tilde{S}) = \sum_{(x,y)\in\tilde{E}} \|x - y\| \geq l \cdot KC_{k-1}(P)$. Therefore,

$$\pi(P) \equiv \max_{S\subset P, |S|=k} \mathrm{div}(S) \geq l \cdot KC_{k-1}(P). \tag{5}$$

Let $S' \subset P$ be a set of size $k$. Let $E', E^*$ be sets of $l$ pairs of points in $S'$ that realize $\mathrm{div}(G(S'))$ and $\mathrm{div}(S')$, respectively.

$$\mathrm{cost}(G(S')) \equiv \mathrm{div}(G(S')) \equiv \sum_{(x,y)\in E'} \|Gx - Gy\|$$

since $\mathrm{div}$ is a minimization problem,

$$\leq \sum_{(x,y)\in E^*} \|Gx - Gy\|$$

by Lemma A.6,

$$\leq \sum_{(x,y)\in E^*} (1+\epsilon)\|x - y\| + \epsilon KC_{k-1}(P)$$
$$\leq (1+\epsilon)\,\mathrm{div}(S') + \epsilon l \cdot KC_{k-1}(P)$$

by Equation (5),

$$\leq (1+\epsilon)\,\mathrm{div}(S') + \epsilon\pi(P).$$

The other direction that $\mathrm{div}(G(S')) \geq (1 - 2\epsilon)\,\mathrm{div}(S) - \varepsilon\pi(P)$ follows by the same arguments. Rescaling $\epsilon$ concludes the proof. $\square$

To prove Lemma A.6, we use the following, which easily follows from (Gupta et al., 2003) (see also the full version of (Jiang et al., 2024)).

**Lemma A.7.** *For every $P \subset \mathbb{R}^d, 0 < \epsilon < 1$ and $k \in \mathbb{N}$, there exists an $\epsilon KC_k(P)$-net of $P$ whose size is $\leq k(2/\epsilon)^{\mathrm{ddim}(P)}$.*

*Proof of Lemma A.6.* Let $N$ be an $\frac{\epsilon}{20}KC_k(P)$-net of $P$ of size $\leq k(40/\epsilon)^{\mathrm{ddim}(P)}$ given by Lemma A.7. For a suitable target dimension $t = O(\epsilon^{-2}\log|N|)$, the following holds: First, by the JL Lemma, with high probability, all the pairwise distances for points in $N$ are preserved up to a factor OF $1 + \epsilon$. Additionally, for every $y \in N$, we have by Lemma 1.2 that with probability $1 - e^{-ct}$, every $x \in P \cup B(y, \frac{\epsilon}{20}KC_k(P))$ satisfies $\|G(x-y)\| \leq 6\frac{\epsilon}{20}KC_k(P)$. Thus, by a union bound, this holds simultaneously for all $y \in N$ with probability at least $1 - e^{-ct}|N| \geq \frac{9}{10}$. Suppose these events hold.

Consider two points $x_1, x_2 \in P$, and let $y_1, y_2 \in N$ be their nearest net-points. Then, by triangle inequality,

$$\|Gx_1 - Gx_2\|$$
$$\leq \|Gx_1 - Gy_1\| + \|Gx_2 - Gy_2\| + \|Gy_1 - Gy_2\|$$

since $\|G(x_i - y_i)\| \leq 6\frac{\epsilon}{20}KC_k(P)$,

$$\leq 12\frac{\epsilon}{20}KC_k(P) + (1+\epsilon)\|y_1 - y_2\|$$

by triangle inequality,

$$\leq 12\frac{\epsilon}{20}KC_k(P) + (1+\epsilon)(\|x_1 - y_1\| + \|x_2 - y_2\| + \|x_1 - x_2\|)$$

since $\|x_i - y_i\| \leq \frac{\epsilon}{20} KC_k$,

$$\leq (14 + 2\epsilon)\frac{\epsilon}{20} KC_k(P) + (1 + \epsilon)\|x_1 - x_2\|$$

and since $\epsilon < 1$,

$$\leq \epsilon KC_k(P) + (1 + \epsilon)\|x_1 - x_2\|.$$

The other direction, that $\|Gx_1 - Gx_2\| \geq (1 - \epsilon)\|x_1 - x_2\| - \epsilon KC_k(P)$, follows by the same arguments. This concludes the proof of Lemma A.6. □

## B. Problems with Large Optimal Value

In this section we show that a large class of optimization problems, including many diversity measures, which have a 'large' optimal value can be preserved via dimensionality reduction in a black-box way. We consider optimization problem of the form "pick the best subset of size $k$ to maximize the 'diversity' from each point to the set" (formalized below).

**Theorem B.1.** *Consider an optimization problem of the following form: given $(P, k, f)$ as input where $P \subset \mathbb{R}^d$, $k \in \mathbb{N}$, and $f : \mathbb{R}^k \to \mathbb{R}$ satisfying $|f(x) - f(y)| \leq L\|x - y\|_\infty$, let*

$$\mathrm{opt}(P) = \max_{S=\{s_1,\ldots,s_k\}\subseteq P, |S|=k} \sum_{p\in P} F(p, S),$$

*where $F(p, S) = f(v_p)$ with $v_p \in \mathbb{R}^k$ with $(v_p)_i = \|p - s_i\|$ for all $i$. Let a Gaussian JL map $G \in \mathbb{R}^{t\times d}$ with suitable $t = O(\varepsilon^{-2}\lambda \log((L + 1)/\varepsilon))$. For every $P \subset \mathbb{R}^d$ with $\mathrm{ddim}(P) = \lambda$, with probability at least $2/3$,*

$$|\mathrm{opt}(P) - \mathrm{opt}(G(P))| \leq O(\varepsilon|P| \cdot \mathrm{diameter}(P)).$$

*Proof.* Set $\Delta = \mathrm{diameter}(P)$. Note that for any set of points $P$, the 1-center cost is $\Omega(\Delta)$. Thus, the $k = 1$ case of Lemma A.6 implies that

$$\Pr\left(\forall x, y \in P, \|Gx - Gy\| \in \|x - y\| + \varepsilon\Delta/100\right) \geq 2/3$$

if we project to $O(\epsilon^{-2} \cdot \lambda \log(1/\varepsilon))$ dimensions (note we can replace multiplicative error with additive error since the additive error is proportional to the diameter). We condition on this event.

Now let $S$ and $S'$ be the maximizing sets for $\mathrm{OPT}(P)$ and $\mathrm{OPT}(G(P))$ respectively. Note that $S'$ is a random variable since it depends on $G$. We will explicitly specify which set of $k$ points is being used to evaluate $f$ and in what space (the original or projected space).

From above, we have that

$$|\|p - s_i\| - \|Gp - Gs_i\|| \leq \varepsilon\Delta/100,$$

for all points $p$ and $s_i$ so it follows that

$$\left|\sum_{p\in P} F(p, S) - \sum_{p\in P} F(Gp, G(S))\right| \leq L \cdot |P| \cdot \varepsilon\Delta/100$$

since $f$ is assumed to satisfy the $\ell_\infty$-Lipschitz condition. Similarly, we have

$$\left|\sum_{p\in P} F(Gp, G(S')) - \sum_{p\in P} F(p, S')\right| \leq L \cdot |P| \cdot \varepsilon\Delta/100.$$

But by definition of optimality, we know

$$\sum_{p\in P} F(p, S) \geq \sum_{p\in P} F(p, S')$$

$$\geq \sum_{p\in P} F(Gp, G(S')) - L \cdot |P| \cdot \varepsilon\Delta/100$$

and similarly

$$\sum_{p \in P} F(Gp, G(S')) \geq \sum_{p \in P} F(Gp, G(S))$$

$$\geq \sum_{p \in P} F(p, S) - L \cdot |P| \cdot \varepsilon\Delta/100,$$

so we have

$$|\text{OPT}(P) - \text{OPT}(G(P))|$$

$$= \left| \sum_{p \in P} F(p, S) - \sum_{p \in P} F(Gp, G(S')) \right|$$

$$\leq L \cdot |P| \cdot \varepsilon\Delta/50.$$

The first part of the theorem follows by rescaling $\varepsilon$, and the 'moreover' part follows by the same arguments. $\square$

While Theorem B.1 is stated in quite general terms, there are many natural choices of functions $f$. We give a non-exhaustive list below:

- $f(x) = \|x\|_p$ which generalizes the Max $k$-coverage diversity measure (which corresponds to $p = \infty$). Note that in the context of Theorem B.1, Max $k$-coverage picks out the largest distance from an input $p$ to the set $S$, whereas the $\ell_p$ formulation combines all distances in a smooth way.

- $f(x) = \sum_i x_i$, which is a special case of the choice above, which in the context of Theorem B.1 averages the distance from every input point to distances to all points in $S$.

- $f(x) = \text{median}(x_i)$, which can be thought of as picking the 'typical' distance from an input point to the set $S$, which maybe more robust to outliers.

In many cases of $f$ stated above, it is true that $\text{OPT}(P) = \Omega(|P| \cdot \text{diameter}(P))$, leading to the following corollary.

**Corollary B.2.** *Consider the setting of Theorem B.1. If* $\text{opt}(P) = \Omega(|P| \cdot \text{diameter}(P))$ *then we can achieve*

$$|\text{opt}(P) - \text{opt}(G(P))| \leq \varepsilon \cdot \text{opt}(P)$$

*with probability at least* $2/3$.

*Remark* B.3. This result extends to preserving approximate solutions by the same arguments.

We highlight two cases where Corollary B.2 holds:

- The maximum degree of the geometric graph defined by all pairwise distances between points in $P$ under dimensionality reduction. This corresponds to $(P, 1, \|x\|_2)$ in Theorem B.1, where $L = 1$.

- The Max $k$-coverage diversity measure which also satisfies $L = 1$ in Theorem B.1. This generalizes the maximum degree example, which is the $k = 1$ case.

**Corollary B.4.** *Consider the maximum $k$-coverage problem:*

$$\text{opt}(P) = \max_{S \subseteq P, |S| = k} \sum_{p \in P} \max_{s \in S} \|p - s\|.$$

*Let a Gaussian JL map* $G \in \mathbb{R}^{t \times d}$ *with suitable* $t = O(\varepsilon^{-2}\lambda \log(1/\varepsilon))$. *For every set* $P \subset \mathbb{R}^d$ *with doubling dimension* $\lambda$, *with probability at least* $2/3$,

$$|\text{opt}(P) - \text{opt}(G(P))| \leq \varepsilon \cdot \text{opt}(P).$$

*Moreover, every* $(1 + \varepsilon)$-*approximate solution of* $\text{opt}(G(P))$ *is a* $(1 + O(\varepsilon))$-*approximate solution of* $\text{opt}(P)$.

Additionally, since the maximum degree of the graph is a special case of a spanning tree, by the same arguments we have the following for maximum spanning tree.

**Theorem B.5.** *Under the settings of Corollary B.4, with probability at least $2/3$, the maximum spanning tree of $G(P)$ is a $(1 + \varepsilon)$-approximation of the maximum spanning tree of $P$.*

Moreover, the dependence in the doubling dimension is tight for these problems, essentially because the optimum value is closely tied to the diameter of the set, so by Lemma A.4, we get the following.

**Theorem B.6.** *Let $n \in N$, there exists a set $P$ of $n$ points, such that for a Gaussian JL map $G$ onto dimension $t$, the cost of maximum spanning tree, maximum degree, and maximum $k$-coverage is distorted by factor $\Omega(\sqrt{\frac{\log n}{t}})$.*

## C. Proofs in Section 2: Max Matching and Max-TSP for Doubling Sets

**Theorem 2.3.** *Let $0 < \epsilon < 1$ and $d, \lambda, k \in \mathbb{N}$, and Gaussian JL map $G \in \mathbb{R}^{t \times d}$ with suitable $t = O(\epsilon^{-2}k^2\lambda \log \frac{k}{\epsilon})$. Then for every set $P \subset \mathbb{R}^d$ with $\mathrm{ddim}(P) = \lambda$, with probability at least $2/3$, a $(1 + \varepsilon)$-approximation to the maximum $k$-hypermatching of $G(P)$ is a $(1 + O(\varepsilon))$-approximation to the maximum $k$-hypermatching of $P$.*

*Proof.* We consider Gaussian JL map as in Lemma 2.2, but with $\varepsilon' = O(\frac{\varepsilon}{k})$. As in the proof of Theorem 2.1, it is immediate that w.h.p., $\mathrm{opt}(G(P)) \geq (1 - \varepsilon)\mathrm{opt}(P)$. Therefore, we focus on proving the other direction.

Consider an optimal $k$-hypermatching of $G(P)$. We can decompose it to $m = O(k)$ matchings $M_1, \ldots, M_m$. Thus, by Lemma 2.2,

$$
\begin{aligned}
\mathrm{opt}(G(P)) &= \sum_{i=1}^{m} \mathrm{cost}(G(M_i)) \\
&\leq \sum_{i=1}^{m} \mathrm{cost}(M_i) + \varepsilon' \, \mathrm{opt}_{\text{max-match}}(P) \\
&\leq \mathrm{opt}(P) + \varepsilon' m \, \mathrm{opt}_{\text{max-match}}(P) \\
&\leq (1 + O(\varepsilon'k)) \, \mathrm{opt}(P) = (1 + \varepsilon) \, \mathrm{opt}(P).
\end{aligned}
$$

$\square$

## D. Proofs in Section 3: $\sqrt{2}$ Approximation Lower Bound

First we begin with some helpful probability statements.

**Lemma D.1.** *Suppose $x \sim \mathcal{N}(0, I_t)$ and let $\gamma = O(1)$. We have*

$$
\Pr(\|x\|_2 \leq \gamma\sqrt{t}) \geq \exp(-O(t \log(1/\gamma))).
$$

*Proof.* Note that if $|x_i| \leq \gamma$ for all $i$ then we must have $\|x\|_2 \leq \gamma\sqrt{t}$ so we lower bound the probability that $|x_i| \leq \gamma$ for all $i$. From the density function of a Gaussian, we know that $\Pr(|x_i| \leq \gamma) = \Theta(\gamma)$, so it follows that

$$
\Pr(\forall i, \ |x_i| \leq \gamma) \geq \Theta(\gamma)^t \geq \exp(-O(t \log(1/\gamma))). \qquad \square
$$

The following is a straightforward corollary of Lemma D.1.

**Corollary D.2.** *Let $\gamma \in (0, 1), t \leq \frac{\log(n)}{C \log(1/\gamma)}$ for a sufficiently large constant $C > 0$. If $x \sim \mathcal{N}(0, I_t)$ then*

$$
\Pr(\|x\|_2 \leq \gamma\sqrt{t}) \geq n^{-O(1/C)}.
$$

**Lemma D.3.** *Suppose $x \sim \mathcal{N}(0, I_t), y \sim \mathcal{N}(0, I_{t-1})$ and fix $\tau > \sqrt{2}$. There exists an absolute constant $C' > 0$ such that*

$$
\Pr(\|y\|_2 \leq \tau) \leq C'^t \cdot \Pr\left(\|x\|_2 \leq \frac{\tau}{\sqrt{2}}\right).
$$

*Proof.* We can decompose

$$\Pr(\|y\|_2 \leq \tau) = \Pr(\|y\|_2 \leq \tau/\sqrt{2}) + \Pr(\tau/\sqrt{2} \leq \|y\|_2 \leq \tau).$$

Note that the former is at least

$$\Pr\left(\|y\|_2 \leq \tfrac{\tau}{\sqrt{2}}\right) \geq \frac{\exp(-\tau^2/4)}{T_{t-1}} \cdot \mathrm{Vol}(B_{t-1}(0, \tau/\sqrt{2})),$$

where $B_{t-1}(0, \tau/\sqrt{2})$ is a centered ball of radius $\tau/\sqrt{2}$ in $\mathbb{R}^{t-1}$ and $T_{t-1}$ is the normalization constant of the Gaussian density function. On the other hand, if $\|y\|_2 \geq \tau/\sqrt{2}$, we can bound

$$\Pr(\tau/\sqrt{2} \leq \|y\|_2 \leq \tau) \leq \frac{\exp(-\tau^2/4)}{T_{t-1}} \cdot \mathrm{Vol}(B_{t-1}(0, \tau))$$

$$\leq \frac{C'^t \exp(-\tau^2/4)}{T_{t-1}} \cdot \mathrm{Vol}(B_{t-1}(0, \tau/\sqrt{2})) \qquad (6)$$

for some sufficiently large constant $C'$. Thus it suffices to compare $\Pr(\|y\|_2 \leq \tau/\sqrt{2})$ and $\Pr(\|x\|_2 \leq \tau/\sqrt{2})$. However, both $\|y\|_2^2$ and $\|x\|_2^2$ are chi-squared random variables (with $\|x\|_2^2$ having one more degree of freedom), and from their density function, it suffices to show

$$2^t \int_0^{\tau/\sqrt{2}} x^{t/2-1} e^{-x/2} \, dx \geq \int_0^{\tau/\sqrt{2}} x^{(t-1)/2-1} e^{-x/2} \, dx.$$

(Note that we have ignored the normalization constant since the ratio of the larger to the smaller can easily be seen to be bounded by $\exp(O(t))$.)

Indeed, point wise, the function $x^{t/2-1} > x^{(t-1)/2-1}$ for all $x > 1$. To handle $x \leq 1$, we have $\int_0^1 x^k \, dx = \frac{1}{k+1}$ so the $2^t$ multiplication term implies the left integral above is also larger over $[0, 1]$. Altogether, combining $\Pr(\|y\|_2 \leq \tau/\sqrt{2}) \leq \exp(O(t)) \Pr(\|x\|_2 \leq \tau/\sqrt{2})$ with Equation 6 completes the proof. $\square$

**Lemma D.4.** *Let $\varepsilon \in (0, 1)$ and $S_1, \cdots, S_n$ be identically distributed and possibly correlated indicator variables satisfying*

- *$\forall i, \mathbf{E}[S_i] = p$,*

- *$p \geq 1/n^{0.1}$,*

- *$\forall k \geq 2$ and distinct indices $i_1, \ldots, i_k, \mathbf{E}[S_{i_1} S_{i_2} \ldots S_{i_k}] \leq (n^{0.001} p)^k$, and*

- *$n \geq \Omega(1/\varepsilon^2)$.*

*Let $S = \sum_{i=1}^n S_i$. We have*

$$\Pr(|S - \mathbf{E}[S]| \geq \varepsilon \mathbf{E}[S]) \leq \frac{1}{n^{99}}.$$

*Proof.* Fix $m \geq 2$ an even integer. We first bound the moment $\mathbf{E}[S^m]$. Note that for all $i, j \geq 1$, $S_i^j = S_i$ since our variables are indicators. Let $1 \leq k \leq m$. Since our random variables are also identical, by symmetry, we only have to understand the number of terms of the form $\mathbf{E}[S_1 \ldots S_k]$ that we get in the multinomial expansion. First, we can pick these $k$ indices in $\binom{n}{k}$ ways. Then by classic stars and bars, there are at most $\binom{m+k-1}{k-1} \leq \binom{m+k}{k}$ ways to assign these $k$ indices all the $m$ powers. This gives us

$$\mathbf{E}[S^m] \leq \sum_{k=1}^m \binom{n}{k} \cdot \binom{m+k}{k} \cdot (n^{0.001} p)^k$$

$$\leq \sum_{k=1}^m \left(\frac{en}{k}\right)^k \cdot \left(\frac{2em}{k}\right)^k \cdot (n^{0.001} p)^k.$$

Since $S$ is non-negative, we know (e.g. see (Nil, 2021)) that

$$\mathbf{E}[|S - \mathbf{E}[S]|^m] \leq \mathbf{E}[S^m],$$

which by Markov's inequality implies

$$\Pr(|S - \mathbf{E}[S] \geq \lambda) \leq \frac{\mathbf{E}[|S - \mathbf{E}[S]|^m]}{\lambda^m} \leq \frac{\mathbf{E}[S^m]}{\lambda^m}.$$

Setting $\lambda = \varepsilon\mathbf{E}[S] = \varepsilon np$ and using our estimate above,

$$\Pr(|S - \mathbf{E}[S] \geq \varepsilon\mathbf{E}[S])$$
$$\leq \sum_{k=1}^m \left(\frac{en}{k}\right)^k \cdot \left(\frac{2em}{k}\right)^k \cdot \frac{(n^{0.001}p)^k}{(\varepsilon np)^m}$$
$$= \sum_{k=1}^m \left(\frac{2e^2m}{k^2}\right)^k \cdot \frac{n^{0.001k}}{n^{m-k}p^{m-k}\varepsilon^m}$$
$$\leq \sum_{k=1}^m \left(\frac{2e^2m}{k^2}\right)^k \cdot \frac{1}{n^{0.9(m-k)-0.001k}\varepsilon^m} \qquad\qquad \text{since } p \geq n^{0.1}.$$

Set $m = n^{0.99}$ (assuming $n^{0.99}$ is an even integer. We omit floor/ceiling signs and a rounding up or down by 1 to ensure evenness, since both are inconsequential). We claim that for any $k \leq m$,

$$\left(\frac{2e^2m}{k^2}\right)^k \cdot \frac{1}{n^{0.9(m-k)-0.001k}\varepsilon^m} \leq \frac{1}{n^{100}}.$$

This is true if and only if

$$n^{100}(2e^2m)^k \leq k^{2k} \cdot n^{0.9(m-k)-0.001k}\varepsilon^m$$
$$\Longleftrightarrow n^{100/k}(2e^2m) \leq k^2 \cdot n^{0.9(m/k-1)-0.001}\varepsilon^{m/k}$$
$$= k^2(n^{0.9}\varepsilon)^{m/k}/n^{0.901}.$$

Since $n \geq 1/\varepsilon^2$, we have $n^{0.9}\varepsilon \geq n^{0.4}$, and for sufficiently large $n$, we have $2e^2m \leq n^{0.999}$, so it suffices to show

$$n^{100/k+1.9} \leq k^2 n^{0.4m/k}$$
$$\Longleftrightarrow \log n\left(\frac{100}{k} + 1.9\right) \leq 2\log(k) + \frac{0.4m}{k} \cdot \log n.$$

If $0.4m/k \geq 100/k + 1.9$ then we are done. Otherwise, we have $.4m/k \leq 100/k + 1.9 \leq 100/k + 2 \implies k \geq .4m - 100 \geq n^{0.98}$ for sufficiently large $n$. Then $100/k + 1.9 \leq 1.95$ for sufficiently large $n$ and we have

$$\log n\left(\frac{100}{k} + 1.9\right) \leq 1.95\log n$$

and $2\log(k) \geq 2 \cdot (0.98)\log n \geq 1.95\log n$, and we are done. Putting everything together, gives us

$$\Pr(|S - \mathbf{E}[S] \geq \varepsilon\mathbf{E}[S]|) \leq \frac{m}{n^{100}} \leq \frac{1}{n^{99}}$$

for sufficiently large $n$, as desired. $\qquad\square$

**Lemma 3.3.** *Let $\varepsilon \in (0,1), n = \Omega(\varepsilon^{-2})$ and $\frac{C\log(1/\varepsilon)}{\varepsilon^2} \leq t \leq \frac{\log(n)}{C\log(1/\varepsilon)}$ for a sufficiently large constant $C > 0$. Let $x_1, \cdots, x_{n/2}$ and $y_1, \cdots, y_{n/2}$ be i.i.d. draws from $\mathcal{N}(0, I_t)$. Consider a bipartite graph $H = (A, B)$ with bipartition $A = \{x_1, \cdots, x_{n/2}\}, B = \{y_1, \cdots, y_{n/2}\}$. Put an edge between $x_i \in A$ and $y_j \in B$ if $\|x_i + y_j\|_2 \leq \varepsilon\sqrt{t}$. With failure probability at most $1/n^{99}$, we have*

$$|\deg(x_1) - \mathbf{E}[\deg(x_1)]| \leq \varepsilon\mathbf{E}[\deg(x_1)].$$

*Proof.* Ultimately our goal is to use the concentration inequality developed in Lemma D.4. Towards this, define the indicator variables $S_i = \mathbf{1}\{\|x_1 + y_i\|_2 \le \varepsilon\sqrt{t}\}$. The degree of $x_1$ is simply the sum of the $S_i$ variables. Note that $S_i$'s are identically distributed but not independent of each other since they all depend on $x_1$. Furthermore, we know that $\forall i, \mathbf{E}[S_i] = p \ge 1/n^{0.1}$ from Corollary D.2 by setting $C$ to be large enough in the definition of $t$. Thus we have checked the first two conditions of Lemma D.4.

The most interesting condition to check is the third hypothesis. For any distinct indices $i_1, \cdots, i_k$,

$$\mathbf{E}[S_{i_1} \cdots S_{i_k}] = \Pr(\forall i_1, \ldots, i_k, \mathbf{1}\{\|x_1 + y_{i_k}\|_2 \le \varepsilon\sqrt{t}\}).$$

We first transform this the event $S_{i_1} \cdots S_{i_k}$ into an equivalent but easier to handle event. Let $U$ be the orthogonal matrix that sends $x_1$ to a scalar multiple of $e_1$, the first basis vector. Note that $U$ is a random matrix that depends on $x_1$ but not on any of the other of the Gaussians that we have drawn. By considering the density function of a Gaussian, we know that $Uy_1, \cdots, Uy_{n/2}$ are i.i.d. Gaussians $z_1, \cdots, z_{n/2}$, again all drawn from $\mathcal{N}(0, I_t)$ (this fact is also referred to as the 'rotational invariance' of Gaussians). Since $U$ is an orthogonal matrix, we always have $\|x_1 + y_i\|_2 = \|Ux_1 + Uy_i\|_2$, so

$$
\begin{aligned}
S_i &= \mathbf{1}[\|x_1 + y_i\|_2 \le \varepsilon\sqrt{t}] \\
&= \mathbf{1}[\|Ux_1 + Uy_i\|_2 \le \varepsilon\sqrt{t}] = \mathbf{1}[\|Ux_1 + z_i\|_2 \le \varepsilon\sqrt{t}],
\end{aligned}
$$

where we again note that $Ux_1$ is a multiple of the vector $e_1$. Now the crux is that for $\|x_1 + y_i\|_2 \le \varepsilon\sqrt{t}$ to hold, it must also be the case that $\|Ux_1 + z_i\|_2 \le \varepsilon\sqrt{t}$, which implies that $\|\tilde{z}_i\|_2 \le \varepsilon\sqrt{t}$ must also hold, where $\tilde{z}_i \in \mathbb{R}^{t-1}$ is the vector where we simply remove the first coordinate of $z_i$ ($\|Ux_1 + z_i\|_2^2$ is a sum of positive terms so if the sum is bounded, any partial sum must also be bounded by the same quantity). Thus,

$$
\begin{aligned}
&\Pr(\forall i_1, \ldots, i_k, \mathbf{1}\{\|x_1 + y_{i_k}\|_2 \le \varepsilon\sqrt{t}\}) \\
&\le \Pr(\forall i_1, \ldots, i_k, \mathbf{1}\{\|\tilde{z}_{i_k}\|_2 \le \varepsilon\sqrt{t}\}).
\end{aligned}
$$

However, the latter probability consists of independent events since the $z_i$'s are i.i.d. Gaussians $N(0, I_{t-1})$. Thus,

$$
\begin{aligned}
&\Pr(\forall i_1, \ldots, i_k, \mathbf{1}\{\|x_1 + y_{i_k}\|_2 \le \varepsilon\sqrt{t}\}) \\
&\le \Pr(\mathbf{1}\{\|\tilde{z}_1\|_2 \le \varepsilon\sqrt{t}\})^k.
\end{aligned}
$$

Since $t = \Omega(1/\varepsilon^2)$ (and thus $\varepsilon\sqrt{t} = \Omega(1)$), Lemma D.3 implies that

$$\Pr(\|\tilde{z}_1\|_2 \le \varepsilon\sqrt{t}) \le C'^t \Pr(\|x_1 + y_{i_1}\| \le \varepsilon\sqrt{t}) = C'^t p$$

for a absolute constant $C' > 0$. Now by setting $C$ in the definition of $t$ to be sufficiently large, we see that $C'^t p \le n^{0.001} p$, satisfying the third condition of Lemma D.4. Thus all the hypothesis of Lemma D.4 hold and we are done. $\square$

**Lemma 3.2.** *Let* $\varepsilon \in (0,1), t \ge C\frac{\log(1/\varepsilon)}{\varepsilon^2}$ *for a sufficiently large constant* $C > 0$. *Let* $x_1, \cdots, x_n$ *be i.i.d. draws from* $\mathcal{N}(0, I_t)$. *With probability* $\ge 1 - \exp(-\Omega(n\varepsilon^2))$, *at least* $1 - \varepsilon$ *fraction of the* $x_i$ *satisfy* $\|x_i\|_2/\sqrt{t} \in 1 \pm \varepsilon$.

*Proof.* From standard concentration bounds for chi-squared variables, (see (Wainwright, 2019)), we know that any fixed $x_i$ satisfies $\|x_i\|_2/\sqrt{t} \in 1 \pm \varepsilon$ with probability at least say $1 - \varepsilon^{100}$. Since the $x_i$ are independent, a standard Chernoff bound implies that at least a $1 - \varepsilon/100$ $x_i$ satisfy $\|x_i\|_2/\sqrt{t} \in 1 \pm \varepsilon$ except with failure probability at most $\exp(-\Omega(n\varepsilon^2))$. $\square$

**Lemma 3.4.** *Consider the setting of Lemma 3.3. If* $C$ *is a sufficiently large constant then* $H$ *has a matching of size* $\ge n/2(1 - \varepsilon)$ *with probability at least* $1 - 1/n^{50}$.

*Proof of Lemma 3.4.* First we show that with high probability, all degrees of $H$ are tightly concentrated. Indeed, using Lemma 3.3 and a union bound over all the $O(n)$ vertices implies that with probability at least $1 - 1/n^{50}$, all vertices have degrees within $1 \pm \varepsilon$ of their expected degree. Condition on this event and set $\Delta = \mathbf{E}[\deg(x_1)]$. For every $(i, j) \in E$, let $z_{ij} = ((1 + \varepsilon)\Delta)^{-1}$. We can verify that for any $x_i \in A$,

$$\sum_{j \in R} z_{ij} = \frac{\deg(x_i)}{(1 + \varepsilon)\Delta} \le 1$$

since we know that $\deg(x_i) \leq (1+\varepsilon)\Delta$ by our conditioning. Thus all the constraints on the vertices in $A$ are satisfied and similarly we can check that all the constraints on the vertices in $B$ are also satisfied. Thus our assignment is feasible and it has total value at least

$$\sum_{(i,j)\in E} z_{ij} = \sum_{i\in A} \sum_{j\in B|(i,j)\in E} \frac{1}{(1+\varepsilon)\Delta}$$
$$\geq \sum_{i\in A} \frac{1-\varepsilon}{1+\varepsilon} \geq |A|(1-O(\varepsilon)).$$

The lemma follows. $\square$

**Theorem 3.5.** *Let $\varepsilon \in (0,1)$, $n = \Omega(1/\varepsilon^3)$ and $P = \{e_1, \cdots, e_n\}$ be the standard basis vectors in $\mathbb{R}^n$. Let $G$ be a Gaussian JL map onto $t$ dimensions. If $\frac{C\log(1/\varepsilon)}{\varepsilon^2} \leq t \leq \frac{\log(n)}{C\log(1/\varepsilon)}$ for a sufficiently large constant $C$, then with probability $\geq 1 - \exp(-\Omega(n\varepsilon^3)) - 1/n^{25}$, we have*

$$\mathrm{opt}_{\mathrm{max\text{-}tsp}}(GP) \geq \left(\sqrt{2} - \varepsilon\right) \cdot \mathrm{opt}_{\mathrm{max\text{-}tsp}}(P).$$

*Proof.* Split the basis vectors into $k = 1/\varepsilon$ sets of $\varepsilon n$ points each. Label the sets $X_1, \cdots, X_{1/\varepsilon}$. By Theorem 3.1, for every consecutive pair $X_i, X_{i+1}$, there exists a matching in the projected space of total weight at least $\varepsilon n(2 - \varepsilon)$ between the pair (with the appropriate failure probability coming from Theorem 3.1). We condition on the event that such a matching exists *for all* consecutive pairs in the projected space (this event happens with probability at least $1 - \exp(-\Omega(n\varepsilon^3)) - 1/n^{25}$ by a union bound).

Given this, we explicitly demonstrate a large weight tour in the projected space. First by following the matchings across all pairs of sets, we have $\varepsilon n$ edge disjoint paths, each with $1/\varepsilon$ edges. Each path has total weight at least $(2 - \varepsilon)/\varepsilon$ by our conditioning, so the total collection has weight at least $n(2 - \varepsilon)$. Connect the endpoints of these paths to form a tour. The extra added connections can only increase the cost of the tour. Thus, we have demonstrated one possible tour in the projected space of cost at least $n(2 - \varepsilon)$. However, any tour in the original space, including the optimal maximum cost tour, has total weight $n\sqrt{2}$, since all distances are $\sqrt{2}$, proving the lower bound as desired. $\square$

## E. $2$-Approximation for Max TSP

We believe that maximum TSP behaves similarly to maximum matching, and a Gaussian JL map yields a $(\sqrt{2} + \varepsilon)$-approximation for maximum TSP with high probability. Such a result would be true assuming that for every set $P \subset \mathbb{R}^d$, there is tour that is a Tverberg graph. However, this is an open question about Tverberg graphs that is still unproven (Pirahmad et al., 2024). We are still able to prove an unconditional bound, of $(2 + \varepsilon)$-approximation.

**Theorem E.1.** *Let $0 < \varepsilon < 1$, $d \in \mathbb{N}$ and a Gaussian JL map $G \in \mathbb{R}^{t\times d}$ for suitable $t = O(\varepsilon^{-2} \log \frac{1}{\varepsilon})$. For every $P \subset \mathbb{R}^d$, with probability at least $2/3$, we have that $\mathrm{opt}_{\mathrm{max\text{-}tsp}}(G(P))$ is a $(2 + \varepsilon)$-approximation of $\mathrm{opt}_{\mathrm{max\text{-}tsp}}(P)$.*

To prove this theorem, we use the following.

**Lemma E.2** ((Indyk, 1999)). *For every set $P \subset \mathbb{R}^d$, the cost of a uniformly random tour is a $(2 + \varepsilon)$-approximation of $\mathrm{opt}_{\mathrm{max\text{-}tsp}}(P)$ with probability $\Omega(\varepsilon)$.*

*Proof of Theorem E.1.* Consider a suitable number $O(\varepsilon^{-1})$ of uniformly random tours. After applying $G$, these are still uniformly random tours. Choose the number of random tours so with probability $2/3$, one of them is a $(2+\varepsilon)$-approximation of $\mathrm{opt}_{\mathrm{max\text{-}tsp}}(G(P))$. By our choice of the target dimension and a union bound, with probability at least $2/3$, the cost of all of these tours is preserved up to factor $1 + \varepsilon$. This concludes the proof. $\square$

## F. Additional Experimental Results

See Table 2 for speedups obtained by dimensionality reduction for the problems we study, as well as Figures 3 to 6 for omitted figures from Section 5.

| Problem | Dataset | Speedup by Computing in Projected Space |
|---|---|---|
| Maximum-Matching | MNIST | 11.41x |
| | CIFAR | 75.07x |
| | Basis Vectors | 10.98x |
| Remote Clique | MNIST | 6.01x |
| | CIFAR | 38.14x |
| | Basis Vectors | 7.50x |
| Maximum Coverage | MNIST | 13.73x |
| | CIFAR | 121.34x |
| | Basis Vectors | 17.37x |

*Table 2.* Speedups obtained by projecting the point sets onto dimension 20, compared to optimizing in the original ambient dimension. For Remote Clique and Max Coverage, we set the value of $k = 10$, although the same qualitative results hold for any $k$. We average over 10 trials.

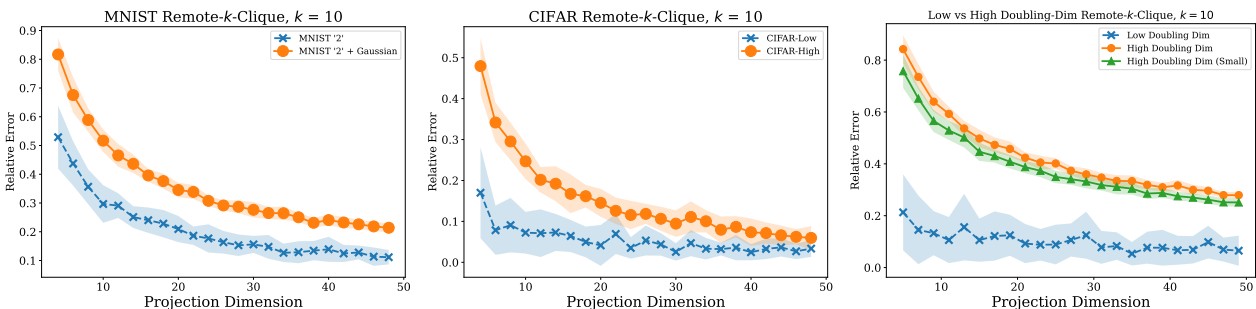

*Figure 3.* Relative error versus projection dimension for remote-$k$-clique. We set $k = 10$ here (see Figure 4 for $k = 20$).

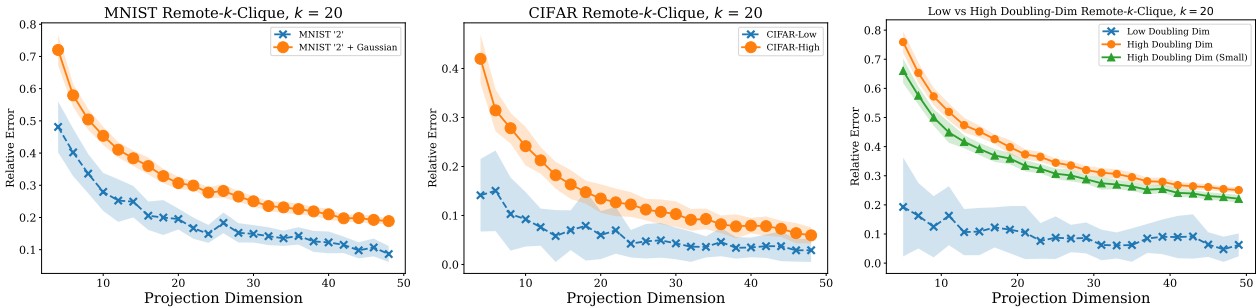

*Figure 4.* Relative error versus projection dimension for remote-$k$-clique. We set $k = 20$.

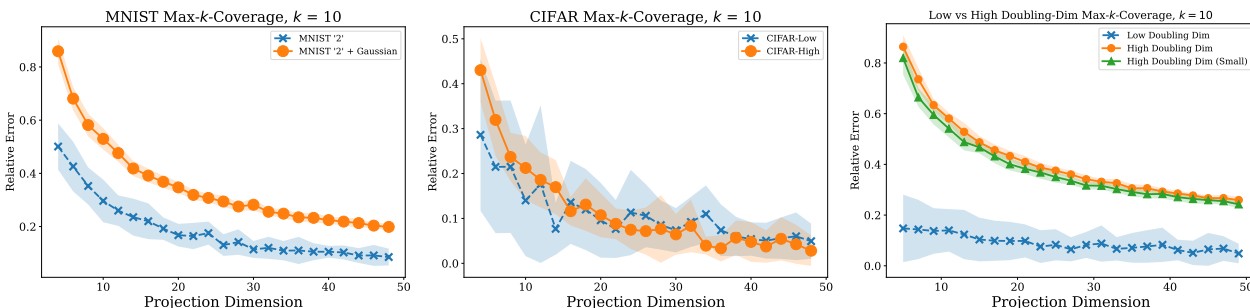

*Figure 5.* Relative error versus projection dimension for max-coverage. We set $k = 10$.

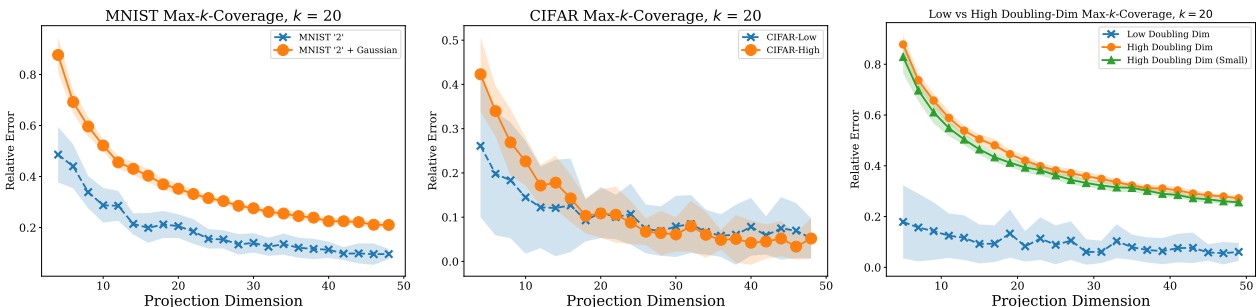

*Figure 6.* Relative error versus projection dimension for max-coverage. We set $k = 20$.

