# OpenReview forum: "Randomized Dimensionality Reduction for Euclidean Maximization and Diversity Measures"
_ICML.cc/2025/Conference — ICML 2025 poster_

### Official Review · Reviewer_jiNf · 2025-03-11

**Overall Recommendation:** 3

**Summary:**

The paper studies several network design problems, including maximum weight matching, maximum TSP, and subgraph diversity on Euclidean doubling space.
In brief, the paper shows the Gaussian JL dimensionality reduction with $O(\lambda \log{(1/\epsilon)} / \epsilon^2)$ dimensions where $\lambda$ is the doubling dimension on the Euclidean space, suffices to achieve $(1 + \epsilon)$-approximation for these problems. The paper also develops lower bounds that show the dependence on $\log{(1/\epsilon)} / \epsilon^2)$ is essential for $(\sqrt{2} - \epsilon)$-approximations.

Previously, similar theoretical results (e.g., reduced dimensions dependent on $\lambda$) exist on approximate near neighbor search (Indyk & Naor 07) and clustering (Narayanan et al. 21, Jiang et al. 24).
Though it studies several related problems, the main upper bound result of the maximum weight matching problem (Lemma 2.2), i.e., the reduced dimensions of $O(\lambda / \epsilon^2)$ suffices to achieve $(1 + \epsilon)$-approximation, is the key tool to show similar results on other problems, including max TSP, max k-hyper-matching, max spanning tree, max k-coverage on Table 1.

**Claims And Evidence:**

Unfortunately, I do not have expertise on these computational problems and the difficulty on showing their approximation so I could not judge the significance of the contributions of this work.

However, I found an issue while trying to understand the proofs of Theorem 2.1.

The max match found on $G(P)$ differs from that found on $ P $ as the random projection matrix $ G $ distorts the pairwise distance within $(1 \pm \epsilon)$ error (say).
Hence, I do not understand how $opt(G(P)) \geq cost(G(S))$ given that $S$ is the optimal solution in $P$ but $G(S)$ is not the optimal solution of $G(P)$ (Line 120).

**Essential References Not Discussed:**

None

**Experimental Designs Or Analyses:**

The empirical designs are quite straighforward and the size of data is small, $n = 1000$.
Since max-matching solver has high complexity in $n$ (NP-hard in geometric space?), I wonder on the signifcance of the reduced dimensionality method vs. the fast approximation solutions of max-matching problem that run in $poly(n)$.

**Methods And Evaluation Criteria:**

The empirical results of max matching compares the quality and running time of the max matching solver on original data vs reduced dimension data. That makes sense since the contribution of the paper show that we can achieve similar results on reduced dimensional space.

**Other Comments Or Suggestions:**

As several theoretical results in the paper are derived from the maximum matching problem in Euclidean space, defining the max match problem and its $(1 + \epsilon)$ approximation would give better to readers.

Lemma 2.4: a data set $P$ with radius $r$? $P$ covered by the ball of radius $r$.
Theorem 3.5: $GP$ should be $G(P)$
Theorem 4.1: $g: R^d \mapsto R$ should be $g: R^d \mapsto R^t$

One of the highlight contributions is showing the existence of a randomized dimensionality reduction for diversity maximization, why would this result be placed in the appendix?

**Other Strengths And Weaknesses:**

I lean on weak reject as I am uncertain on the signifcance of the contribution. I will engage the discussion to understand the problem better and will change the decision based on other reviewers' comments.

---------------

I change the score to Weak Accept after reading the rebuttal messages.

**Questions For Authors:**

Q1) What are the definitions of max matches and its $(1 + \epsilon)$-approximation?

**Relation To Broader Scientific Literature:**

I do not know since I am not familiar with these problems and its hardness to achieve $(1 + \epsilon)$-approximation.

**Theoretical Claims:**

I have not checked the proofs since I am not expertise in these problem.

---

> ### Author Rebuttal · Authors · 2025-03-31
>
> We thank the reviewer for their careful reading and comments. We address the weaknesses they mentioned below.
>
> **Significance of the contribution:** We give an example for obtaining a significantly faster algorithm for estimating weighted matchings, but similar examples exist for all the other problems considered here. Given a weighted graph with $n$ nodes and $m$ edges, one can estimate the size of a matching up to a $(1+\varepsilon)$ factor in time $O(m+n)$, ignoring lower order terms [1]. In Euclidean spaces, this is always $O(n^2)$, as all edges exist and computing all of these edges takes time $O(n^2d)$. This is clearly not a linear time algorithm as the input has only size $O(nd)$. A fast alternative using coresets exists [2], but it requires time $O(n\cdot\exp(d))$, which for worst case instances does not offer an improvement even when using previous dimension reduction bounds. Specifically, our lower bound shows that we can only replace $d$ with $O(\log n)$ in the worst case. However, if the doubling dimension is constant, we can use the coreset in combination with our new dimension reduction bounds to obtain a linear time algorithm, which to the best of our knowledge was not previously known.
>
> In general, we prove that for a wide range of optimization problems, any dependency on $d$ may be replaced with a dependency on the doubling dimension. This almost always speeds up an algorithm, as the running time of the random projection is inexpensive. But it is particularly useful if the algorithm runs in time $\exp(d)$, as illustrated for the matching example above. Such examples are common in literature for the other problems considered here as well, due to the proverbial curse of dimensionality. The running time also gets reduced significantly in practice. Our experiments show that the runtimes are reduced by up to $10-100$x as shown in Table 2 in the appendix, and the reduction in solution quality is small.
>
> - [1] R. Duan and S. Petite. Linear-Time Approximation for Maximum Weight Matching. J. ACM 2014.
> - [2] G. Frahling and C. Sohler. Coresets in Dynamic Geometric Data Streams. STOC 2005.
>
>
> **Definition for max-matching:** A matching is a set of pairs of points, with no point belonging to more than one pair; the cost of a matching is the sum of distances between the pair of points; and a maximum matching is a matching with the maximum cost. A (1+eps)-approximate matching is a matching whose cost is a (1+eps)-approximation to the cost of a max-matching. In other words, it is an approximation whose cost is approximately as large as the best possible cost.
>
> **Relation to broader scientific literature:** We would like to point out that many papers related to the JL lemma and random projection based dimensionality reduction have recently appeared in top ML venues. See below for a small selection.
>
> - Beyond Worst-Case Dimensionality Reduction for Sparse Vectors. ICLR 25
> - MUVERA: Multi-Vector Retrieval via Fixed Dimensional Encodings NeurIPS 24
> - Sparse Dimensionality Reduction Revisited. ICML 24
> - Dynamic Metric Embedding into lp Space. ICML 24
> - Simple, Scalable and Effective Clustering via One-Dimensional Projections. NeurIPS 23
> - Fast Optimal Locally Private Mean Estimation via Random Projections. NeurIPS 23
> - Dimensionality Reduction for General KDE Mode Finding. ICML 23
> - Dimensionality reduction for Wasserstein barycenter. NeurIPS 21
> - Randomized dimensionality reduction for facility location and single-linkage clustering. ICML 21
> - Dimensionality Reduction for the Sum-of-Distances Metric. ICML 21
>
> **Explaining proof of Theorem 2.1:** We consider maximum matching, so the cost of an optimal solution is larger than the cost of any other solution. Opt$(G(P))$ is the optimal solution in the projected space by definition, so it must be larger than the solution given by $G(S)$ where $S$ is the optimum in the original dimension.
>
>
> **Regarding placing the proof for diversity maximization in the appendix:** Ideally, all proofs should be in the main body. However, due to the page limit, we chose to provide the entire proof for the max-matching problem, rather than providing fragmented proofs for all the claims.
>
> We believe we have addressed your main concern about the significance of the contribution. Please let us know if you have any further concerns or questions; we are very happy to provide further clarifications!
>
> Many thanks,
>
> The authors

---

> > ### Comment · Reviewer_jiNf · 2025-04-02
> >
> > Thank the authors for detailed feedback. I have increased my score to Weak Accept as I am familiar a few papers related to this work.

---

### Official Review · Reviewer_6YCZ · 2025-03-12

**Overall Recommendation:** 4

**Summary:**

The paper studies randomized dimensionality reduction for a range of the Euclidean optimization problems, including max-matching,
max-spanning tree, max TSP, max k-coverage, and subgraph diversity. In particular, the paper relates the target dimension to the doubling dimension $\lambda_X$ of the dataset $X$ and shows that $O(\lambda_X)$ suffices to approximately preserve the near-optimal solution. The paper also provides a lower dimension bound for a $\sqrt{2}$ approximation. Finally, the paper also gives an empirical evaluation that shows the speed-up of the proposed dimensionality reduction and demonstrates that the effects of doubling dimension is an empirically observable phenomenon which can be quantitatively measured.

**Claims And Evidence:**

Yes.

**Essential References Not Discussed:**

N/A

**Experimental Designs Or Analyses:**

The experimental design looks reasonable to me.

**Methods And Evaluation Criteria:**

Yes.

**Other Comments Or Suggestions:**

See the next question.

**Other Strengths And Weaknesses:**

Strengths:

- The paper is technically solid. The paper relates the target dimension to the doubling dimension $\lambda_X$ and gives a careful analysis, which is interesting to me.

- The paper gives a detailed empirical evaluation that demonstrates the power of the proposed dimensionality reduction and the effects of doubling dimension is an empirically observable phenomenon which can be quantitatively measured.

- The organization of the paper is generally good and easy to follow.

Weaknesses:

I currently do not see any other major weakness of the paper.

**Questions For Authors:**

In Table 1, the paper claims that they prove a $\lambda$ dimension lower bound for these optimization problems. However, it seems to me they only give an $O(\log n)$ lower bound, which corresponds to the special case ($\lambda = \log n$) but not for a general range of $\lambda$?

**Relation To Broader Scientific Literature:**

See "Summary".

**Theoretical Claims:**

The proof in the main body looks correct to me, but I did not carefully check the proof in the appendix. I have some question about the claim made in Table 1 (for details, see the below question).

---

> ### Author Rebuttal · Authors · 2025-03-31
>
> > However, it seems to me they only give an lower bound, which corresponds to the special case (lambda = log n) but not for a general range of lambda?
>
> We thank the reviewer for their careful reading and comments.
>
> The lower bound can be extended to general $\lambda$ as follows. For every $\lambda$, consider the pointset that consists of the $2^\lambda$ first basis vectors, and duplicate each point $n/2^\lambda$ times. If you are concerned with having a set with multiplicities, you can move the copies slightly along one axis (this will not affect the doubling dimension).

---

### Official Review · Reviewer_74V2 · 2025-03-13

**Overall Recommendation:** 4

**Summary:**

This paper studies randomized dimensionality reduction for geometric optimization problems such as max-matching, max-TSP, and max-spanning tree. It introduces a novel approach where the reduction is based on the doubling dimension of the dataset instead of the dataset size. The authors prove that reducing the dimension to O(λX), where λX is the dataset's doubling dimension, is sufficient to preserve the value of near-optimal solutions. They provide both theoretical proofs and experimental results to validate this claim. The experiments show that this method maintains solution quality while significantly improving computational efficiency.

**Claims And Evidence:**

Some theoretical claims, particularly on the optimality gap after dimensionality reduction, lack detailed derivations.
The experimental validation is limited to specific datasets (e.g., MNIST, CIFAR). It is unclear how well the method generalizes to other domains, such as NLP or structured data.
The paper does not directly compute the doubling dimension (λX) but estimates it through experimental trends. This indirect approach makes it unclear how to determine λX efficiently in real-world applications.

**Essential References Not Discussed:**

The paper does not compare well with manifold learning techniques , which also reduce dimensions while preserving geometric structures.

**Experimental Designs Or Analyses:**

1.The paper mainly uses image-based datasets, which may not reflect the challenges of high-dimensional sparse data.
2.The paper does not provide a direct computation method for λX but relies on experimental behavior to infer its impact. It is unclear how well this estimation method applies to other types of data.
3.The paper mainly compares against JL transforms but does not benchmark against other adaptive dimensionality reduction techniques (e.g., PCA).

**Methods And Evaluation Criteria:**

The method relies on estimating the doubling dimension, but it is not clear how to precisely compute λX for arbitrary datasets. In some cases, especially for high-sparsity data (e.g., NLP embeddings), estimating λX may be difficult or computationally expensive.

**Other Comments Or Suggestions:**

Please see the above weakness

**Other Strengths And Weaknesses:**

Strengths
1.Theoretical novelty: The paper introduces an innovative way to determine the target dimension using the doubling dimension.
2.Computational efficiency: The method shows a clear advantage in reducing computational costs.
3.General applicability: It applies to a variety of geometric optimization problems.
Weaknesses
1.Limited empirical validation: The experiments do not cover enough diverse datasets.
2.Unclear computation of doubling dimension: The paper does not provide a direct method to compute λX, which limits its practical use.
3.Missing comparisons: Other adaptive dimensionality reduction techniques (e.g., PCA, deep learning-based methods) are not compared.

**Questions For Authors:**

Please see the above weakness

**Relation To Broader Scientific Literature:**

The paper contributes to the intersection of dimensionality reduction and combinatorial optimization. It builds on the Johnson-Lindenstrauss lemma, extending it to optimization problems.

**Theoretical Claims:**

The method assumes that estimating λX is feasible, but does not provide a practical way to compute it efficiently.
It would be helpful to discuss whether similar guarantees hold in non-Euclidean spaces.

---

> ### Author Rebuttal · Authors · 2025-03-31
>
> We thank the reviewer for their careful reading and comments. We address the weaknesses they mentioned below.
>
> **1) Datasets:** We focused on datasets that have been used in prior empirical studies on diversity maximization and dimensionality reduction (Tenenbaum et al., 2000; Naeem et al. 2020). Furthermore, we believe our data is quite high dimensional. E.g. the Resnet embeddings we use are in dimension > 6000.
>
> As for sparse NLP datasets, **we performed a new experiment** on maximum matching on TF-IDF embeddings for the 20 newsgroup dataset. These resulted in extremely sparse vectors with dimension > 170,000 (TF-IDF is based on word frequencies). We selected a set of 2000 vectors. The average sparsity was 100. As seen in the figure in this anonymous link (https://ibb.co/2YsR6kXS), the qualitative behaviour is the same as in our submission. By just projecting to 500 dimensions (~ 0.2% of the original dimension), we can preserve the max-matching cost up to relative error < 5%. In fact, since our bounds depend on the doubling dimension (always at most $O(\log n)$), their effect becomes increasingly pronounced as the ambient dimension of the dataset increases.
>
> **2) Comparison to manifold learning:**
> Our paper focuses on proving worst-case theoretical guarantees. We do this by using data-oblivious maps based on the Johnson-Lindenstrauss (JL) Lemma.
>
> In contrast, the methods suggested by the reviewer have no such theoretical guarantees for Euclidean distance, and can perform very poorly. This is well known, e.g. see [Ref 1] where it is shown theoretically and empirically that JL has better performance over PCA and Isomap heuristics. The underlying intuition is that manifold learning (such as Isomap) considers data points that are in a lower dimensional manifold embedded in a high dimensional space. Their goal is to recover the low dimensional manifold by approximating the geodesic distance along the unknown manifold. However, the geodesic distance could be very far from Euclidean. This deviates from our setting where we aim to approximate high dimensional Euclidean distances.
>
> For concreteness, here is a simple example where PCA catastrophically fails: consider the dataset X consisting of all basis vectors and negations. Weight the first $n/2$ basis vectors and their negatives by a factor of $2$. This has doubling dimension $O(\log(|X|))$ and JL guarantees that all pairwise distances (and thus all the optimization problems we consider) is preserved up to $1\pm \epsilon$ factor when projected to $O(\log(|X|)/\epsilon^2)$ dimensions. However, for any $k < n/2$, the top $k$ PCA directions align with the first $k$ basis vectors. When we project onto them, this maps all other basis vectors to $0$, so all information about them is lost, and max-matching on this PCA projected dataset has large distortion.
>
> [Ref 1]: Dimensionality reduction: theoretical perspective on practical measures. NeurIPS 19.
>
> We also refer to the response to Reviewer jiNF for many works related to the JL lemma that have recently appeared in top ML venues.
>
> **3) Doubling dimension:** The way JL transforms are applied is by determining a desired target dimension and performing the projection. In practice, users will use as many dimensions as are affordable and for many problems, worst case bounds are significantly larger than the dimensions that end up being used. Naturally, we are interested in understanding why this phenomenon occurs. An explanation is that data sets have low “intrinsic dimension”, which the doubling dimension captures and models. This is a popular approach in the literature which has led to development of several practically efficient algorithms (e.g. Cover Tree by Beygelzimer et al.). Our real-data experiments exemplify this, and demonstrate that data sets with low ``intrinsic dimension'' can be projected to very low dimension without significantly reducing the accuracy. This is why we selected real data sets (e.g. MNIST 2) that were studied in prominent works studying embeddings of data sets with low intrinsic dimension (Tenenbaum et al, Science'00).
>
> We emphasize that the algorithm does not have to know the doubling dimension and as long as the target dimension is larger than the doubling dimension, it is always guaranteed to succeed with high probability. Works in this line of research typically only have to show that a given target dimension is sufficient, for a specific task, as the algorithm itself is oblivious to the dataset. This sets random projections apart from PCA and manifold learning and similar methods that are computationally expensive, have to know properties of the data set, and typically perform very poorly as metric embedding algorithms (see earlier example on PCA). If it is nevertheless desired by the user, we could compute an approximation of the doubling dimension in linear time (see Sect. 9 in https://arxiv.org/pdf/cs/0409057), but the efficiency of our method does not rely on this.

---

### Decision · Program_Chairs · 2025-05-01

**Decision:**

Accept (poster)

**Comment:**

All three reviews of this work [74V2,6YCZ,jiNf] leaned towards acceptance with two Accept ratings and one Weak accept rating.

The reviewers appreciated several aspects of the work:
- The way of using doubling dimension to determine the target dimension for dimension reduction was considered innovative [74V2] and interesting [6YCZ]
- The advantage in computational efficiency was appreciated [74V2]
- The detailed empirical evaluation was appreciated [74V2]
- Applicability to several geometric optimization problems was appreciated [74V2]
- The paper was considered well organized and easy to follow [6YCZ]

However, several concerns were also raised:

- The lack of detailed derivation for some theoretical claims was criticized [74V2]
- Proof of one lower bound was noted to apply only to a special case [6YCZ]
- Limited experimental validation on specific (mainly image-based) datasets was criticized, leaving concerns about domain generalizability [74V2]
- One reviewer was unclear of the difficulty of showing approximations for the computational problems and hence on significance of the contributions [jiNf]
- An estimate of the doubling dimension was needed, but it was unclear how to efficiently compute it for arbitrary datasets in real applications [74V2]
- Discussion of guarantees for non-Euclidean spaces was desired [74V2]
- Limited comparison methods were criticized and benchmarking against more adaptive dimension reduction techniques like PCA was desired [74V2]; similarly, comparison to manifold learning techniques was desired [74V2]
- Additional experiments on larger/more diverse datasets were desired [74V2]
- An issue of clarity was noted for proofs of Theorem 2.1 [jiNf]
- There was concern whether the reduced-dimensional method yields significant improvement versus fast approximation solutions [jiNf]
- Better definition of the maximum matching problem and its (1+epsilon) approximation were desired [jiNf]

Authors provided rebuttals where they addressed some points, including providing discussion of the significance of their contributions (yielding a faster algorithm for weighted matchings), giving a definition of max-matching, and running a new experiment on the 20 newsgroups dataset. They also gave an argument against manifold learning methods and PCA, some discussion of the doubling dimension and whether it needs to be estimated well, and a brief sketch of a proof generalizability extension for the concern of [6YCZ] and a brief additional explanation for Theorem 2.1 for [j1Nf]. The feedback led to score improvements for [74V2,jiNf].

Overall, I find that some of the concerns identified by the reviewers above remain for the paper, particularly due to limited experimental validation. However, like the reviewers I appreciate the theoretical contributions. Thus, although the paper is a borderline case, ultimately I lean slightly towards acceptance.